# Stability and Generalization of Kernel Clustering: From Single Kernel to Multiple Kernel

**Weixuan Liang**[1]     **Xinwang Liu**[1,∗]    **Yong Liu**[2,3]     **Sihang Zhou**[4]     **Jun-Jie Huang**[1]

**Siwei Wang**[1]          **Jiyuan Liu**[1]          **Yi Zhang**[1]          **En Zhu**[1]

[1]School of Computer, National University of Defense Technology (NUDT), Changsha, China
[2]Gaoling School of Artificial Intelligence, Renmin University of China, Beijing, China
[3]Beijing Key Laboratory of Big Data Management and Analysis Methods, Beijing, China
[4]College of Intelligence Science and Technology, NUDT, Changsha, China

## Abstract

Multiple kernel clustering (MKC) is an important research topic that has been widely studied for decades. However, current methods still face two problems: inefficient when handling out-of-sample data points and lack of theoretical study of the stability and generalization of clustering. In this paper, we propose a novel method that can efficiently compute the embedding of out-of-sample data with a solid generalization guarantee. Specifically, we approximate the eigen functions of the integral operator associated with the linear combination of base kernel functions to construct low-dimensional embeddings of out-of-sample points for efficient multiple kernel clustering. In addition, we, for the first time, theoretically study the stability of clustering algorithms and prove that the single-view version of the proposed method has uniform stability as $\mathcal{O}\left(Kn^{-3/2}\right)$ and establish an upper bound of excess risk as $\widetilde{\mathcal{O}}\left(Kn^{-3/2} + n^{-1/2}\right)$, where $K$ is the cluster number and $n$ is the number of samples. We then extend the theoretical results to multiple kernel scenarios and find that the stability of MKC depends on kernel weights. As an example, we apply our method to a novel MKC algorithm termed SimpleMKKM and derive the upper bound of its excess clustering risk, which is tighter than the current results. Extensive experimental results validate the effectiveness and efficiency of the proposed method.

## 1   Introduction

Multiple kernel clustering (MKC) [33] aims to learn an optimal kernel from a set of pre-specified kernel matrices for high-quality partition. These methods usually assume that the optimal kernel matrix is a linear combination of the pre-specified base kernels. By jointly optimizing kernel weights and a clustering indicator matrix, MKC improves the clustering performance of single-view kernel $k$-means (KKM). In recent years, MKC algorithms have been extensively studied. Among these works, Lu *et al.* [23] propose to change the MKC framework into the form of kernel alignment. Gönen *et al.* [9] propose to employ a localized method to obtain more sample-specific information from the data. Recently, Liu *et al.* [20] propose a parameter-free MKC algorithm that adopts a novel min-max optimization framework and achieves state-of-the-art performance on a wide range of applications. Although impressive improvements have been made by the existing methods, there

---

∗Corresponding author: `xinwangliu@nudt.edu.cn`

36th Conference on Neural Information Processing Systems (NeurIPS 2022).

still lacks an effective and computationally efficient mechanism to handle out-of-sample problems. This makes them inefficient in both space and computational consumption for large-scale unseen samples. More specifically, since all existing methods need to conduct eigen-decomposition over the matrix consisting of all visible samples as an intermediate operation, the pre-extracted information is blocked for the new coming samples. This makes the algorithms computationally expensive to handle large-scale datasets.

To improve kernel clustering scalability, methods such as Nyström [30] approximation and random Fourier feature (RFF) [3] are proposed. These two methods acquire the non-linear feature of samples in real space by approximating the kernel matrix. However, the Nyström method can't be implemented directly on out-of-sample points of MKC directly. The shortcoming of RFF is also obvious, i.e., the dimension of the learned embedding is comparably large such that the subsequent clustering process is time-consuming. More seriously, because it's difficult to bound the difference of the kernel weights before and after the approximation, these two methods are rarely implemented on MKC algorithms. In this paper, we propose a general method with a solid generalization guarantee. More precisely, we find that the main complexity is caused by recomputing eigenvectors for the embedding of new coming samples. To this end, through the study of the integral operator associated with the kernel functions, we propose to perform an eigen-decomposition approximation [27] for MKC. Specifically, we learn eigen functions from the empirical optimal kernel generated from base kernels and then use these eigen functions to calculate the approximation embedding for the out-of-sample data points. Then, a standard $k$-means algorithm can be performed on the learned embedding for clustering.

To theoretically analyze our algorithm, we propose a novel stability evaluation criterion for multiple kernel clustering and establish the generalization bound accordingly. Algorithmic stability [2] is a key property to estimate the generalization bound of learning algorithms. The stability-based technique [6, 14] is widely used in analyzing the generalization ability of learning algorithms. However, existing literature lacks research on the stability of clustering algorithms. Unlike supervised learning, which possesses labels as a deterministic standard, the risk function of clustering tasks is highly correlated with uncertain clustering centroids. Therefore, it is hard to analyze the algorithm stability for clustering. To address the issue, we start from single-view kernel $k$-means with a pairwise learning framework. We show that the proposed method has uniform stability as $\mathcal{O}\left(Kn^{-3/2}\right)$, where $K$ is the number of clusters and $n$ is the number of samples. Consequently, the excess risk can be upper bounded by $\widetilde{\mathcal{O}}\left(Kn^{-3/2} + n^{-1/2}\right)^2$. We then extend the generalization bound analysis from single kernel clustering to multiple kernel clustering and prove that the stability of MKC is related to the learned kernel weights. As an instantiation, we use the above results to analyze the simple multiple kernel $k$-means (SimpleMKKM)[20] and derive a generalization bound as $\widetilde{\mathcal{O}}\left((m+1)Kn^{-3/2} + n^{-1/2}\right)$, where $m$ is the number of kernels. As far as we know, this is the tightest bound for MKC algorithms. Finally, we conduct numerical experiments to verify the effectiveness and efficiency of the proposed method.

Our contributions are three-fold and can be summarized as follows:

1) We propose an efficient method to enable MKC algorithms to handle out-of-sample problems.

2) We successfully apply algorithmic stability to derive the generalization bound of the clustering algorithm for the first time. We study the stability of SimpleMKKM with our method and derive its excess risk bound.

3) By comprehensive numerical experiments, the experimental results show that the proposed algorithm is effective and efficient.

The rest of the paper is organized as follows. Section 2 introduces the notations used in this paper. Section 3 discusses the related work. Section 4 describes the proposed method. Section 5 analyzes the stability and generalization of KKM on the single kernel and then extends the proposed method to multiple kernel clustering. The experimental results are reported in Section 6. Finally, the paper is concluded in Section 7. All proofs are placed in the appendix due to the limited space.

---

[2]$\widetilde{\mathcal{O}}$ hides logarithmic terms.

## 2 Notations

To avoid confusion with mathematical notations, the uppercase and lowercase letters in bold denote matrices and vectors, respectively. For example, $\mathbf{A}$ is a matrix, and $\mathbf{a}$ is a vector. The component of them is denoted by $A_{ij}$ or $a_i$. Let us denote with $\mathcal{X}$ the sample space and with $\mathbb{P}$ the corresponding data distribution. The training dataset $S = \{\mathbf{x}_i\}_{i=1}^n \subset \mathcal{X}$ is drawn i.i.d. from $\mathbb{P}$. The empirical distribution $\mathbb{P}_n$ is defined as $\mathbb{P}_n(\mathbf{x}) = \frac{1}{n}$ if $\mathbf{x} \in S$, otherwise 0. We assume that kernel function $k(\cdot, \cdot) : \mathcal{X} \times \mathcal{X} \to \mathbb{R}$ is a positive-definite and conjugate symmetric function. From [1], we know that there exist a Hilbert space $\mathcal{H}$ and a feature mapping $\phi(\cdot) : \mathcal{X} \to \mathcal{H}$ such that $\forall \mathbf{x}, \mathbf{x}' \in \mathcal{X}$, $k(\mathbf{x}, \mathbf{x}') = \langle \phi(\mathbf{x}), \phi(\mathbf{x}') \rangle_{\mathcal{H}}$. The kernel function $k(\cdot, \cdot)$ used in this paper satisfies that $\forall \mathbf{x}, \mathbf{x}' \in \mathcal{X}$, $k(\mathbf{x}, \mathbf{x}) = k(\mathbf{x}', \mathbf{x}')$. We assume that for any $\mathbf{x} \in \mathcal{X}$, $\|\phi(\mathbf{x})\| \le 1$. Let us denote with $\mathbf{I}_s$ the $s \times s$ identity matrix, and with $\|\cdot\|_\infty$ the infinity norm. For the convenience, the probability measure $\rho$ after differential $d$ is omitted, *i.e.*, $d\mathbf{x}$ is short for $d\rho(\mathbf{x})$.

## 3 Related Work

In this section, we give a brief introduction to pairwise learning and algorithmic stability which are two main contents related to our proofs.

**Pairwise Learning**. Pairwise learning defines loss function with respect to sample pairs. It has been successfully applied to metric learning[31], bipartite ranking [29], AUC maximization [25], and etc. Assume that $\{\mathbf{x}_1, \cdots, \mathbf{x}_n\}$ are $n$ samples drawn independently from some probability measure. Suppose that the learning model is parameterized by $\mathbf{w}$. The empirical behavior of the learning model can then be evaluated by the empirical risk as:

$$\hat{R}(\mathbf{w}) = \frac{1}{n^2} \sum_{i,j=1}^n l(\mathbf{x}_i, \mathbf{x}_j; \mathbf{w}),$$

where $l$ denotes the loss function. Accordingly, the expected risk can be defined as:

$$R(\mathbf{w}) = \iint l(\mathbf{x}, \mathbf{y}; \mathbf{w}) d\mathbf{x} d\mathbf{y}.$$

Among the researches of learning theory, Lei *et al.* [15] establish a sharper generalization bound of pairwise learning, Li *et al.* [17] utilize a pairwise learning framework to verify the generalization ability of clustering algorithms. In [16], Lei *et al.* study the generalization of stochastic gradient descent for pairwise learning. In [4, 17], the authors learn the excess risk bound of clustering in the form of pairwise learning. As mentioned above, in the centroid-based clustering algorithms, it is generally hard to investigate the stability of the samples at the boundary of two centroids. To tackle this problem, we adopt a pairwise learning framework in our analysis.

**Algorithmic Stability**. The analysis of algorithmic stability is a powerful technique for deriving the generalization bound of stable learning algorithms. In the existing literature, uniform stability [2] is the most widely used one but needs the strongest assumptions. A variety of stability notions have been proposed for different algorithms to widen the application range of stability. Kutin *et al.* [14] define strong and weak stability for the algorithms which are not uniformly stable. Locally elastic stability is proposed in [6] to capture the sensitivity of the loss function in the population sense but not in the worst sense. Lei *et al.* [15] provide a notation of uniform stability for pairwise learning. Despite the fact that algorithmic stability has been successfully used in analyzing numerous learning algorithms, no work applies it to clustering algorithms. In this paper, we show that the clustering algorithm we studied is uniformly stable.

## 4 Framework

In this section, we first give a brief introduction to kernel $k$-means and multiple kernel clustering, then introduce the proposed approximation method.

### 4.1 Kernel $k$-means and Multiple Kernel Clustering

Given a training set $S = \{\mathbf{x}_i\}_{i=1}^n$ and a feature map $\phi(\cdot)$, kernel clustering algorithm aims to minimize the following objective *w.r.t.* clustering centroids:

$$\mathcal{W}(\mathbf{C}, \mathbb{P}_n) = \frac{1}{n} \sum_{i=1}^{n} \min_{k \in [K]} \|\phi(\mathbf{x}_i) - \mathbf{c}_k\|^2, \tag{1}$$

where $\mathbf{C} = \{\mathbf{c}_k\}_{k=1}^{K}$ denotes the centroids of $K$ clusters.

A clustering indicator matrix $\mathbf{H} \in \mathbb{R}^{n \times K}$ is defined as follows: if $\mathbf{x}_i \in \mathcal{C}_k$, $h_{ik} = 1/\sqrt{|\mathcal{C}_k|}$, otherwise $h_{ik} = 0$, where $|\mathcal{C}_k|$ denotes the number of samples belong to the $k$-th cluster.

**Kernel $k$-means.** If $\phi(\cdot)$ is defined by some single kernel function $k(\cdot, \cdot)$, then Eq.(1) can be rewritten as:

$$\min_{\mathbf{H}} \frac{1}{n} \mathrm{Tr}(\mathbf{K}_n) - \frac{1}{n} \mathrm{Tr}(\mathbf{K}_n \mathbf{H} \mathbf{H}^\top), \tag{2}$$

where $\mathbf{K}_n$ is the empirical kernel matrix computed by the kernel function $k$ and dataset $S$, *i.e.*, $K_{ij} = k(\mathbf{x}_i, \mathbf{x}_j)$.

The discrete constraint of $\mathbf{H}$ in Eq.(2) can be relaxed as: 1) $h_{ik}$ can take any real number; 2) the orthogonal constraint still holds, *i.e.*, $\mathbf{H}^\top \mathbf{H} = \mathbf{I}_K$. Consequently, the solution to Eq.(2) are the $K$ eigenvectors corresponding to the $K$ largest eigenvalues of kernel matrix $\mathbf{K}_n$. Therefore, $\mathbf{H}$ can be regarded as a new embedding for the training dataset, and standard $k$-means can then be performed on $\mathbf{H}$ for the final clustering. To avoid confusion, in this paper, kernel $k$-means refers to the relaxed version as mentioned above.

**Multiple Kernel Clustering.** In the multiple kernel setting, a sample has $m(m \geq 2)$ feature mappings $\{\phi_p(\mathbf{x})\}_{p=1}^{m}$ associated with $m$ base kernel functions $\{k_p\}_{p=1}^{m}$. As the setting of multiple kernel learning [8], multiple kernel clustering (MKC) usually lets each kernel multiply by a weight $\alpha_p \ (p \in [m])$. These weights satisfy $\sum_{p=1}^{m} \alpha_p = 1$ and $\alpha_p \in [0, 1]$. Thus, the feature mapping of MKC can be represented as:

$$[\alpha_1 \phi_1(\mathbf{x}), ..., \alpha_m \phi_m(\mathbf{x})]^\top.$$

Similar to Eq.(2), Eq.(1) in multiple kernel setting can be reformed as:

$$\min_{\mathbf{H}} \frac{1}{n} \mathrm{Tr}(\mathbf{K}_{\boldsymbol{\alpha}}) - \frac{1}{n} \mathrm{Tr}(\mathbf{K}_{\boldsymbol{\alpha}} \mathbf{H} \mathbf{H}^\top), \tag{3}$$

where $\mathbf{K}_{\boldsymbol{\alpha}} = \sum_{p=1}^{m} \alpha_p^2 \mathbf{K}_p$. MKC aims to learn a set of optimal combination weights. Then, the relaxed kernel $k$-means is performed on $\mathbf{K}_{\boldsymbol{\alpha}}$ for a unified embedding $\mathbf{H}$. Notice that $\mathbf{K}_{\boldsymbol{\alpha}}$ is the kernel matrix of the training dataset. To obtain the embedding of an out-of-sample point, we have to perform eigen-decomposition on the kernel matrix constructed by the training dataset and the new coming data point. Obviously, this process is inefficient.

### 4.2 The Proposed Approximation Method

In our approach, we do not approximate the kernel matrix but the eigen functions of the integral operator associated with the kernel function. First, we give a brief analysis of the spectrum of the integral operator. We start from the risk function of a sample and the expected risk, which are respectively defined as:

$$l(\mathbf{x}, \mathbf{C}) = \min_{k \in [K]} \|\phi(\mathbf{x}) - \mathbf{c}_k\|^2, \quad \mathbb{E}\left[l(\mathbf{x}, \mathbf{C})\right] = \int \min_{k \in [K]} \|\phi(\mathbf{x}) - \mathbf{c}_k\|^2 d\mathbf{x}. \tag{4}$$

Similar to Eq.(2), we reform the risk function of MKC for any two samples $\mathbf{x}, \mathbf{y} \in \mathcal{X}$ as follows:

$$l(\mathbf{x}, \mathbf{y}, H) = k(\mathbf{x}, \mathbf{x}) - \sum_{k=1}^{K} k_{\boldsymbol{\alpha}}(\mathbf{x}, \mathbf{y}) h_k(\mathbf{x}) h_k(\mathbf{y}), \tag{5}$$

where $k_{\boldsymbol{\alpha}}(\cdot, \cdot) = \sum_{p=1}^{m} \alpha_p^2 k_p(\cdot, \cdot)$ denotes the linear combination of base kernel functions, and $H = \{h_k(\mathbf{x})\}_{k=1}^{K}$ denotes the clustering indicator functions which are defined as:

$$h_k(\mathbf{x}) = \begin{cases} 1/\sqrt{\int_{\mathbf{x} \in \mathcal{C}_k} d\mathbf{x}}, & \mathbf{x} \in \mathcal{C}_k, \\ 0, & \mathbf{x} \notin \mathcal{C}_k. \end{cases} \tag{6}$$

---

**Algorithm 1** MKC-AE

---

**Require:** Out-of-sample example $\mathbf{x}$; kernel weights $\{\alpha_p\}_{p=1}^m$, clustering indicator matrix $\mathbf{H}_{\boldsymbol{\alpha}}$ and the largest $K$ eigenvalues $\{\lambda_{k,\boldsymbol{\alpha}}\}_{k=1}^K$ learned by some MKC on some training set; $m$ kernel functions $\{k_p\}_{p=1}^m$; cluster number $K$.

 1: **for** $k = 1 : K$ **do**
 2:     For sample $\mathbf{x}$, compute the $k$-th element of the embedding vector by Eq.(8).
 3: **end for**

**Ensure:** A $K$-dimension approximation embedding for $\mathbf{x}$.

---

The expected clustering risk is then defined as:

$$R(H, \mathbb{P}) = \iint l(\mathbf{x}, \mathbf{y}, H) d\mathbf{x} d\mathbf{y}. \tag{7}$$

The following proposition shows that the expected risk of the proposed pairwise learning framework equals that of the original kernel clustering algorithm with the same cluster centroids. Thus, we can learn the excess risk bound with the proposed framework defined by Eq.(5) instead of the original one defined by Eq.(4).

**Proposition 4.1.** *The expected clustering risk defined by Eq.(7) can be reformed as:*

$$R(H, \mathbb{P}) = \int \min_{k \in [K]} \|\phi(\mathbf{x}) - \mathbf{c}_k\|^2 d\mathbf{x} = \int k(\mathbf{x}, \mathbf{x}) d\mathbf{x} - \sum_{k=1}^K \iint h_k(\mathbf{x}) k_{\boldsymbol{\alpha}}(\mathbf{x}, \mathbf{y}) h_k(\mathbf{y}) d\mathbf{x} d\mathbf{y}.$$

Obviously, $\{h_k(\mathbf{x})\}_{k=1}^K$ are all unit functions and orthogonal to each other in the squared integrable space $\mathcal{L}^2(\mathcal{X}, \rho)$. Similar to Eq.(2), the discrete constraint of these functions can also be relaxed, and the relaxed version is investigated in this paper. Denote that $H^* = \{h_k^*(\mathbf{x})\}_{k=1}^K$ are the $K$ eigen functions corresponding to the $K$ largest eigenvalues of the integral operator $L_k : \mathcal{L}^2(\mathcal{X}, \rho) \to \mathcal{L}^2(\mathcal{X}, \rho)$,

$$(L_k f)(\mathbf{x}) = \int_{\mathcal{X}} k_{\boldsymbol{\alpha}}(\mathbf{x}, \mathbf{y}) f(\mathbf{y}) d\rho(\mathbf{y}).$$

Obviously, $H^*$ is the minimizer of the expected clustering risk defined by Eq.(7). However, $H^*$ cannot be obtained because the data distribution is unknown. Rosasco *et al.* [27] analyze the spectral properties of the integral operator defined by kernel function and its empirical version, i.e., kernel matrix. Inspired by their method, we aim to construct clustering indicator functions from the empirical kernel matrix that can approximate $H^*$. Suppose that the combination weights learned by some MKC algorithms on $S$ are $\{\alpha_p\}_{p=1}^m$, the embedding of $\frac{1}{n}\mathbf{K}_{\boldsymbol{\alpha}}$ is $\mathbf{H}_{\boldsymbol{\alpha}}$ and the largest $K$ eigenvalues are $\{\lambda_{k,\boldsymbol{\alpha}}\}_{k=1}^K$. We construct $K$ functions as follows:

$$\tilde{h}_{k,\boldsymbol{\alpha}}(\mathbf{x}) = \frac{1}{n\lambda_{k,\boldsymbol{\alpha}}} \left( \sum_{i=1}^n h_{ik,\boldsymbol{\alpha}} \left( \sum_{p=1}^m \alpha_p^2 k_p(\mathbf{x}, \mathbf{x}_i) \right) \right), \tag{8}$$

where $k$ ranges in $[K]$ and $h_{ik,\boldsymbol{\alpha}}$ is the $ik$-th element of $\mathbf{H}_{\boldsymbol{\alpha}}$. Then, the approximation embedding for an out-of-sample point $\mathbf{x}$ can be constructed by a $K$-dimensional vector $[\tilde{h}_{1,\boldsymbol{\alpha}}(\mathbf{x}), ..., \tilde{h}_{K,\boldsymbol{\alpha}}(\mathbf{x})]$. The above procedures are listed in Algorithm 1, and we term this algorithm as multiple kernel clustering with approximation embedding (MKC-AE). This method has two noticeable advantages: *1) It can be applied to all MKC algorithms based on embedding learning. 2) It has a theoretical generalization guarantee which will be studied in the following section.*

**The application on large-scale datasets.** The proposed MKC-AE method can be adapted to perform MKC on large-scale datasets. Let us denote with $S_N$ a large-scale dataset consisting of $N$ samples. We first select $n$ samples as landmarks and perform some MKC algorithm on these landmarks. The remaining $N - n$ samples can be regarded as "out-of-sample" examples, and the approximation embedding of these samples can be obtained by the following process. Assume that the optimal kernel matrix of some MKC algorithm on $n$ landmarks is $\frac{1}{n}\mathbf{K}_{n,\boldsymbol{\alpha}}$. The matrix constructed by the $K$ largest eigenvectors of $\frac{1}{n}\mathbf{K}_{n,\boldsymbol{\alpha}}$ is denoted as $\mathbf{H}_n \in \mathbb{R}^{n \times K}$, and the corresponding eigenvalues matrix is $\mathbf{D}_n \in \mathbb{R}^{K \times K}$. Assume that the partial kernel similarity constructed by $N$ samples and $n$

landmarks are $\{\mathbf{P}_l\}_{l=1}^m \subset \mathbb{R}^{N \times n}$, *i.e.*, $[\mathbf{P}_l]_{ij} = k_l(\mathbf{x}_i, \mathbf{x}_j), \forall l \in [m], i \in [N], j \in [n]$. According to Eq.(8), the approximation embedding of $S_N$ can be computed as:

$$\hat{\mathbf{H}}_N = \left( \sum_{l=1}^m \alpha_l^2 \mathbf{P}_l \right) \mathbf{H}_n \mathbf{D}_n^{-1}.$$

Notice that the main complexity of computing the approximation embedding is $\mathcal{O}\left(nmN + nKN\right)$, which is caused by the summation of $\{\mathbf{P}_l\}_{l=1}^m$ and subsequent matrix multiplication. We then give an example to show how the proposed MKC-AE method can accelerate MKC. The time complexity of the MKC algorithm termed SimpleMKKM proposed in [20] is basically cubic with the number of training samples. Performing SimpleMKKM on all $N$ samples will cost $\mathcal{O}\left(N^3\right)$ time, while the proposed MKC-AE method reduces the complexity to $\mathcal{O}\left(n^3 + nmN + nKN\right)$. When $n \ll N$, the complexity is linear with the number of samples $N$. Thus, the proposed algorithm can enable MKC algorithms to be performed on large-scale datasets.

## 5 Theoretical Analysis

As the foundation for the theoretical analysis of multiple kernel clustering, we first analyze the stability of kernel $k$-means (KKM) with our approximation method and then establish the generalization bound.

### 5.1 Stability and Generalization of Clustering on Single Kernel

Notice that there are two intractable problems in the analysis for the stability of kernel clustering: *1) The clustering risk is concerned with centroids. However, the random initialization of clustering centroids makes the stability of kernel clustering hard to be analyzed; 2) To compute the clustering risk of a sample, it needs to find the nearest centroid. As a result, it is difficult to estimate the stability of the samples located at the boundary of two centroids.* In this paper, since the proposed method adopts a pairwise risk function, the above two issues can be avoided. The detailed deduce process is as follows.

In the single kernel setting, let us denote with $\frac{1}{n}\mathbf{K}_n$ the (normalized) empirical kernel matrix constructed by the training set, and suppose that the $K$ largest eigenvalues and the corresponding eigenvectors of $\frac{1}{n}\mathbf{K}_n$ are $\{\lambda_k\}_{k=1}^K$ and $\{\mathbf{h}_k\}_{k=1}^K$, respectively. According to Eq. (8), the approximated clustering indicator functions $\tilde{H} = \{\tilde{h}_k\}_{k=1}^K$ are given by:

$$\tilde{h}_k(\mathbf{x}) = \frac{1}{n\lambda_k} \left( \sum_{i=1}^n h_{ik} k(\mathbf{x}, \mathbf{x}_i) \right), \tag{9}$$

where $h_{ik}$ is the $i$-th component of $\mathbf{h}_k$. We aim to bound the excess clustering risk as:

$$\begin{aligned}
&\mathbb{E}_S R(\tilde{H}, \mathbb{P}) - R(H^*, \mathbb{P}) \\
&= \underbrace{\mathbb{E}_S R(\tilde{H}, \mathbb{P}) - \mathbb{E}_S R(\tilde{H}, \mathbb{P}_n)}_{B} + \underbrace{\mathbb{E}_S R(\tilde{H}, \mathbb{P}_n) - R(H^*, \mathbb{P})}_{C}.
\end{aligned} \tag{10}$$

By the following proposition, we know the second item $C \leq 0$ in Eq.(10).

**Proposition 5.1.** *In Eq.(10),* $\mathbb{E}_S R(\tilde{H}, \mathbb{P}_n) \leq R(H^*, \mathbb{P})$.

To give the upper bound of $B$, we first prove that the approximation method given by Eq.(9) has uniform stability $\mathcal{O}\left(\frac{K}{n\sqrt{n}}\right)$ with respect to the risk function defined by Eq.(5) .

**Theorem 5.2.** *Denote that two training datasets $S, S^i$ have $n$ samples and differ by only one sample[3]. Suppose that the empirical kernel matrix has properties as: 1) there exist two constant $c, c_1$ such that the $K$-th eigenvalue is larger than $1/c$; and 2) the gaps of first $K + 1$ eigenvalues are larger than $c_1$, i.e., $\forall k \in [K], \lambda_k - \lambda_{k+1} \geq c_1 \geq \frac{4}{\sqrt{n}}$. The approximation clustering indicator functions of $S, S^i$*

---

[3]In fact, $S^i$ is a set derived by replacing the $i$-th sample of $S$.

*obtained by Eq.(9) are denoted as $\tilde{H}$ and $\tilde{H}^i$, respectively. Then, for any $\mathbf{x}, \mathbf{y} \in \mathcal{X}$, there exists a constant $c_0 > 0$ such that*

$$|l(\mathbf{x}, \mathbf{y}, \tilde{H}) - l(\mathbf{x}, \mathbf{y}, \tilde{H}^i)| \leq \frac{c_0 K}{n\sqrt{n}}. \tag{11}$$

**Remark 1**. Suppose that the $K$-th eigenvalue of the integral operator associated with kernel function $k(\cdot, \cdot)$ is $\overline{\lambda}_K$. Then, by Proposition 10 in [27], the $K$-th eigenvalue of the corresponding empirical kernel matrix $\lambda_K$ converges to $\overline{\lambda}_K$ when $n$ approaches infinity. When the kernel function is fixed, $\overline{\lambda}_K$ can be considered as a constant. Consider that a sequence of the $K$-th eigenvalue $\lambda_K^n$ with different sample number $n$. Let $\epsilon = \frac{\overline{\lambda}_K}{2}$, then there exists a positive integer $N$ such that $\forall n \geq N$, $|\lambda_K^n - \overline{\lambda}_K| \leq \frac{\overline{\lambda}_K}{2}$. We can let $1/c = \min\{\lambda_k^1, \cdots, \lambda_K^N, \frac{\overline{\lambda}_K}{2}\}$, then the assumption that $\lambda_K \geq 1/c$ holds. Similarly, the assumption about eigen gap of the kernel matrix is related to the eigen gap of the integral operator corresponding to the kernel function, thus appropriate kernel function can make the assumption hold. We conduct experiments to verify that the kernel functions we adopted satisfy the assumptions in Theorem 5.2. The detailed experimental results are reported in Section B.4.

The following theorem [15] is about the generalization risk bound of the stable pairwise learning algorithm.

**Theorem 5.3.** *[15] Assume that algorithm $A$ has $\gamma$-stability. If there exists a constant $M > 0$ such that $|\mathbb{E}_S[l(\mathbf{x}, \mathbf{y}, A(S))]| < M$ holds for all $\mathbf{x}, \mathbf{y} \in \mathcal{X}$. Then for all $\delta \in (0, 1/e)$, the following inequality holds with probability $1 - \delta$:*

$$|R(A(S), \mathbb{P}_n) - R(A(S), \mathbb{P})| \leq 4\gamma + e\left(12\sqrt{2}M\sqrt{\frac{\log(e/\delta)}{n-1}} + 48\sqrt{6}\gamma\lceil\log_2 n\rceil \log\left(\frac{e}{\delta}\right)\right).$$

It is easy to check that the proposed risk function defined by Eq.(5) can be upper bounded by some constant $M$. Then, combining Theorem 5.2 and Theorem 5.3, the proposed method has a generalization risk bound $B$ as $\tilde{\mathcal{O}}\left(\frac{K}{n\sqrt{n}} + \frac{1}{\sqrt{n}}\right)$. As seen, the item $\tilde{\mathcal{O}}\left(\frac{1}{\sqrt{n}}\right)$ masters the whole bound. Notice that this is the first bound for clustering deduced by algorithmic stability, and we can use it to learn the stability and generalization of multiple kernel clustering algorithms. In the future, we will improve this bound by further investigating the generalization theorem about the proposed pairwise learning framework.

## 5.2 Stability and Generalization of Clustering on Multiple Kernel

To study the generalization ability of the proposed MKC-AE method on multiple kernel, we first reveal the relation between the stability of MKC-AE and the learned kernel weights.

**Theorem 5.4.** *Denote that two training datasets $S, S^i$ have $n$ samples and differ by only one sample. Assume that for any kernel weights $\boldsymbol{\alpha}$, the empirical kernel matrix $\frac{1}{n}\mathbf{K}_{\boldsymbol{\alpha}}$ satisfies the same properties in Theorem 5.2. If $\|\boldsymbol{\alpha} - \boldsymbol{\alpha}^i\|_\infty \leq \eta$, there exists a constant $c_0$ such that*

$$|l(\mathbf{x}, \mathbf{y}, \widetilde{H}_{\boldsymbol{\alpha}}) - l(\mathbf{x}, \mathbf{y}, \widetilde{H}_{\boldsymbol{\alpha}^i}^i)| \leq \frac{c_0 m K}{n}\eta + \frac{c_0 K}{n\sqrt{n}},$$

*holds.*

**Remark 2**. Theorem 5.4 shows that if the kernel weights of an MKC algorithm are stable, then its approximation method proposed in Algorithm 1 has uniform stability. Specifically, if $\eta$ in Theorem 5.4 can be bounded by $\mathcal{O}(1/\sqrt{n})$, it can be checked that the risk function has the stability in order of $\mathcal{O}\left(\frac{(m+1)K}{n\sqrt{n}}\right)$. In this regard, for a tighter risk bound, the kernel weights of MKC algorithms should be more stable. This can be a design principle for MKC algorithms.

As an instantiation, we use the above theoretical results to analyze simple multiple kernel $k$-means (SimpleMKKM) [20], which is a state-of-the-art MKC algorithm without any hyper-parameters. We first give a brief introduction and then derive the generalization bound of SimpleMKKM with the proposed approximation method.

In the task of unsupervised learning, finding the optimal hyper-parameters is an open question. SimpleMKKM is proposed as a hyper-parameter-free MKC algorithm and achieves desirable clustering

performance. The variants of SimpleMKKM [19, 22] further boost the clustering performance and widen its application. Specifically, SimpleMKKM aims to solve the following kernel alignment-based optimization problem:

$$\min_{\boldsymbol{\alpha} \in \triangle} F(\boldsymbol{\alpha}), \tag{12}$$

where $F(\boldsymbol{\alpha}) = \max_{\mathbf{H}} \frac{1}{n}\mathrm{Tr}\left(\mathbf{K}_{\boldsymbol{\alpha}}\mathbf{H}\mathbf{H}^{\top}\right)$, s.t. $\mathbf{H}^{\top}\mathbf{H} = \mathbf{I}_K$, and $\triangle = \{\boldsymbol{\alpha} \in \mathbb{R}^m | \sum_{p=1}^m \alpha_p = 1, \alpha_p \geq 0, \forall p \in [m]\}$. $F(\boldsymbol{\alpha})$ in Eq.(12) is proven differentiable, and can be optimized by a reduced gradient algorithm [26]. One can refer to Section A.5 in the appendix for a detailed optimization procedure.

The following theorem indicates that the learned kernel weights of SimpleMKKM are stable with the input samples.

**Theorem 5.5.** *Assume that for any kernel weights $\boldsymbol{\alpha}$, the empirical kernel matrix $\frac{1}{n}\mathbf{K}_{\boldsymbol{\alpha}}$ satisfies the same properties in Theorem 5.2. Denote that the kernel weights obtained by performing SimpleMKKM on $S, S^i$ are respectively $\boldsymbol{\alpha}, \boldsymbol{\beta}$, and the number of iterations is $T$. If in each iteration, the index of the largest element of $\boldsymbol{\alpha}$ is the same as $\boldsymbol{\beta}$, then there exists a constant $c_0 \geq 1$ such that*

$$\|\boldsymbol{\alpha} - \boldsymbol{\beta}\|_{\infty} \leq \frac{Tc_0^T}{\sqrt{n}},$$

*holds.*

By Theorem 5.4 and Theorem 5.5, we have the following corollary. We know that SimpleMKKM with the proposed method has uniform stability, thus the generalization bound can be established.

**Corollary 5.6.** *Assume that for any kernel weights $\boldsymbol{\alpha}$, and the empirical kernel matrix $\frac{1}{n}\mathbf{K}_{\boldsymbol{\alpha}}$ satisfies the same properties in Theorem 5.2. Then, there exists a constant $c_0$ such that*

$$|l(\mathbf{x}, \mathbf{y}, \widetilde{H}_{\boldsymbol{\alpha}}) - l(\mathbf{x}, \mathbf{y}, \widetilde{H}_{\boldsymbol{\alpha}^i}^i)| \leq \frac{c_0(m+1)K}{n\sqrt{n}},$$

*holds.*

**Remark 3**. From the empirical results of Section B.3 in the appendix, SimpleMKKM always converges within several iterations, and thus $T$ can be regarded as a constant. Above all, according to Theorem 5.3, we know that the excess risk of SimpleMKKM with the approximation method can be bounded by $\tilde{\mathcal{O}}\left(\frac{(m+1)K}{n\sqrt{n}} + \frac{1}{\sqrt{n}}\right)$. Our bound is tighter than the existing results $\mathcal{O}(\frac{mK}{n})$ proposed in [20]. As far as we know, this is the first time to study the generalization ability of MKC with stability, and the provided bound is the tightest among existing results of MKC algorithms.

## 6 Experiments

Besides the theoretical analysis, we further validate the effectiveness of the proposed method experimentally. To illustrate that the proposed method can handle out-of-sample problems properly, we first apply the proposed method to SimpleMKKM [20] and compare the clustering accuracy and efficiency of the proposed method and the original SimpleMKKM [20]. To evaluate the scalability of our method, we again apply it to SimpleMKKM and compare it against several state-of-the-art efficient MKC algorithms on four large-scale datasets. The analysis of convergence is reported in Section B.3 of the appendix.

### 6.1 The Comparison with SimpleMKKM

We follow the settings in SimpleMKKM [20] and conduct experiments on 5 benchmark datasets, including *Flo17*, *Flo102*, *DIGIT*, *Cal102* and *Reuters* to evaluate the effectiveness of the proposed algorithm over the out-of-sample problem. The details of the above datasets are listed in Section B.1 of the appendix. We first uniformly select $n$ samples at random from the original dataset, and the remaining $N - n$ samples are regarded as out-of-sample points. We perform SimpleMKKM on the $n$ samples for the related parameters and compute the approximated embedding by Eq.(8). We record the clustering performance of the approximated embedding. For comparison, the results of SimpleMKKM on all $N$ samples are also recorded. In our experiments, $n$ is ranged in $[K, 200, 300, \cdots, 1000]$,

where $K$ is the cluster number. To avoid the influence of randomness, we run the experiments 20 times with the same setting. All experiments are performed on a desktop with Intel(R) Core(TM)-i7-7820X CPU and 64G RAM.

We record the clustering performance of the proposed method (denoted as proposed) and SimpleMKKM (denoted as original) in terms of accuracy (ACC), normalized mutual information (NMI), purity, and execution time in Table 1. The best performance of the proposed method and the corresponding running time are reported among different landmarks. As observed in Table 1, the proposed method is comparable with SimpleMKKM in terms of the clustering performance evaluation indexes. This verifies that the approximated embedding obtained by our method is as effective as the exact embedding. On the other hand, the processing time of the proposed method is much less than that of SimpleMKKM. We can also see that the proposed method achieves a more evident acceleration effect with an increasing number of samples. For example, in *Flo17*, the proposed method works **2.35** times faster than SimpleMKKM, while in *Reuters*, the proposed method is around **238.5** times faster than SimpleMKKM.

Table 1: Experimental results of the proposed method in comparison with the original SimpleMKKM.

|  | Flo17 | | Digit | | CCV | | Flo102 | | Reuters | |
|---|---|---|---|---|---|---|---|---|---|---|
|  | Original | Proposed | Original | Proposed | Original | Proposed | Original | Proposed | Original | Proposed |
| ACC | 59.60 | 60.26 | 90.50 | 91.40 | 22.04 | 21.26 | 42.07 | 41.92 | 45.82 | 45.09 |
| NMI | 57.55 | 58.77 | 83.62 | 84.38 | 18.21 | 17.95 | 58.46 | 58.39 | 27.76 | 26.78 |
| Purity | 60.64 | 61.62 | 90.50 | 91.40 | 25.21 | 25.09 | 48.47 | 47.80 | 53.28 | 52.67 |
| Time | 41.14 | **17.51** | 35.9 | **2.57** | 370.3 | **43.42** | 2026.3 | **140.6** | 3715.1 | **15.58** |

We also study the clustering performance of the proposed method with different numbers of landmarks. Due to limited space, please refer to Section B.2 for details.

## 6.2 Experiments on Large-scale Datasets

As mentioned in Section 4, the proposed method can enable MKC to be applied to large-scale datasets. To verify the efficiency, we evaluate the performance of the proposed algorithm on 4 large-scale datasets, including *NUSWIDE*, *AwA* and *MNIST* and *YtVideo*. The sample numbers of these datasets are all larger than 30,000, and the maximal one is 101,499. The detailed information of these datasets is reported in Section B.1 of the appendix.

For comparison, we conduct experiments on several SOTA large-scale multi-view clustering algorithms, including (1) **scalable and parameter-free multi-view graph clustering (SFMC)** [18] which is a graph-based multi-view clustering algorithm with adaptively selected anchors; (2) **binary multi-view clustering (BMVC)** [32] which encodes each graphical view into a binary form for lower computational complexity; (3) **large-scale multi-view subspace clustering (LMVSC)** [13] which constructs a matrix with selected anchors to reduce the redundant computation of subspace clustering. For the comparison algorithms, we select the best hyper-parameters by grid searching as suggested in the comparison methods. For sufficient landmarks, we set the number of landmarks of the proposed method as 1000.

Table 2: Experimental results of different clustering methods on large-scale datasets. N/A means the corresponding method is out of memory. (The best result in each row is in bold.)

| Dataset | Metric | SFMC | BMVC | LMVSC | Proposed+SimpleMKKM |
|---|---|---|---|---|---|
| NUSWIDE | ACC | 14.02 | 13.83 | 14.50 | **15.24** |
|  | NMI | 3.97 | 12.84 | 11.84 | **14.02** |
|  | Purity | 15.87 | 24.43 | 15.52 | **26.33** |
|  | Time | 268.64 | 28.28 | 73.68 | **20.46** |
| AwA | ACC | 5.26 | 8.66 | 8.68 | **9.65** |
|  | NMI | 2.32 | 11.75 | 9.65 | **11.92** |
|  | Purity | 5.34 | 10.73 | 10.02 | **11.81** |
|  | Time | 3327.17 | 23.91 | 80.49 | **16.31** |
| MNIST | ACC | N/A | 45.99 | 98.97 | **99.09** |
|  | NMI | N/A | 39.86 | 96.86 | **97.17** |
|  | Purity | N/A | 45.99 | 98.97 | **99.09** |
|  | Time | N/A | 26.08 | 46.74 | **6.34** |
| YtVideo | ACC | N/A | **19.41** | 17.25 | 17.35 |
|  | NMI | N/A | **15.80** | 14.08 | 15.24 |
|  | Purity | N/A | 30.78 | **32.25** | 29.58 |
|  | Time | N/A | 61.23 | 161.26 | **53.88** |

We can see from Table 2 that the proposed method outperforms all comparison algorithms in *NUSWIDE*, *AwA* and *MNIST*. For *YtVideo* data, it is a vast dataset, but our algorithm still works and achieves comparable performance. Moreover, the proposed method achieves the fastest processing speed among all comparison methods. Above all, it can be verified that the proposed algorithm is efficient and effective on large-scale datasets.

# 7    Conclusions and Future Work

In this paper, we propose a simple yet effective approximation embedding method for multiple kernel clustering. It can be applied to accelerate the existing MKC algorithms to efficiently obtain the embedding of out-of-sample points, therefore is suitable for clustering large-scale datasets. We, for the first time, theoretically study the stability of clustering algorithms and establish the generalization bound of the proposed method. We first give the learning bound of relaxed KKM with algorithmic stability which is then extended for the analysis of MKC. Finally, the experimental results verify the effectiveness and efficiency of the proposed method.

Although we give some theoretical results about the generalization of clustering algorithms by stability, some assumptions still limit the application range of our theorems, *e.g.*, the assumptions about eigenvalues. In the future, we aim to 1) establish the stability and generalization bounds for a wider range of MKC algorithms based on milder assumptions; 2) determine the optimal number of landmarks through theoretical analysis.

## Acknowledgments and Disclosure of Funding

This work was supported by the National Key R&D Program of China 2020AAA0107100 and the National Natural Science Foundation of China (project no. 61773392, 61872377, 61922088, 61976196, and 62006237).

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
