# A Proofs

## A.1 Proof of Proposition 4.1

*Proof.* Denote the function $I_k : \mathcal{X} \to \{0, 1\}$ as $I_k(\mathbf{x}) = 1$ if $\mathbf{x}$ belongs the $k$-th cluster, and otherwise $I_k(\mathbf{x}) = 0$. Then, we have

$$\int_{\mathbf{x} \in \mathcal{X}} \min_k \|\phi(\mathbf{x}) - \mathbf{c}_k\|^2 d\mathbf{x}$$

$$= \int \min_k \left\| \phi(\mathbf{x}) - \frac{\int \phi(\mathbf{y}) I_k(\mathbf{y}) d\mathbf{y}}{\int I_k(\mathbf{y}) d\mathbf{y}} \right\|^2 d\mathbf{x}$$

$$= \int \min_k \left( k(\mathbf{x}, \mathbf{x}) - 2 \frac{\int k(\mathbf{x}, \mathbf{y}) I_k(\mathbf{y}) d\mathbf{y}}{\int I_k(\mathbf{y}) d\mathbf{y}} + \frac{\iint k(\mathbf{x}, \mathbf{y}) I_k(\mathbf{x}) I_k(\mathbf{y}) d\mathbf{x} d\mathbf{y}}{(\int I_k(\mathbf{y}) d\mathbf{y})^2} \right) d\mathbf{x}$$

$$= \int \sum_{k=1}^{K} I_k(\mathbf{x}) \left( k(\mathbf{x}, \mathbf{x}) - 2 \frac{\int k(\mathbf{x}, \mathbf{y}) I_k(\mathbf{y}) d\mathbf{y}}{\int I_k(\mathbf{y}) d\mathbf{y}} + \frac{\iint k(\mathbf{x}, \mathbf{y}) I_k(\mathbf{x}) I_k(\mathbf{y}) d\mathbf{x} d\mathbf{y}}{(\int I_k(\mathbf{y}) d\mathbf{y})^2} \right) d\mathbf{x}$$

$$= \int \sum_{k=1}^{K} I_k(\mathbf{x}) k(\mathbf{x}, \mathbf{x}) d\mathbf{x} - \sum_{k=1}^{K} \frac{\iint k(\mathbf{x}, \mathbf{y}) I_k(\mathbf{x}) I_k(\mathbf{y}) d\mathbf{x} d\mathbf{y}}{\int I_k(\mathbf{y}) d\mathbf{y}}$$

$$= \int k(\mathbf{x}, \mathbf{x}) d\mathbf{x} - \sum_{k=1}^{K} \frac{\iint k(\mathbf{x}, \mathbf{y}) I_k(\mathbf{x}) I_k(\mathbf{y}) d\mathbf{x} d\mathbf{y}}{\int I_k(\mathbf{y}) d\mathbf{y}}$$

$$= \int k(\mathbf{x}, \mathbf{x}) d\mathbf{x} - \sum_{k=1}^{K} \iint h_k(\mathbf{x}) k(\mathbf{x}, \mathbf{y}) h_k(\mathbf{y}) d\mathbf{x} d\mathbf{y}$$

$$= R(H, \mathbb{P}).$$

The proof is complete.

$\square$

## A.2 Proof of Proposition 5.1

*Proof.* It is easy to check that

$$\mathbb{E}_S \hat{R}(\tilde{H}, \mathbb{P}_n)$$

$$= \mathbb{E}_S \left[ \frac{1}{n^2} \sum_{i,j=1}^{n} \left( k(\mathbf{x}_i, \mathbf{x}_i) - \sum_{k=1}^{K} k(\mathbf{x}_i, \mathbf{x}_j) \tilde{h}_k(\mathbf{x}_i) \tilde{h}_k(\mathbf{x}_j) \right) \right]$$

$$\leq \mathbb{E}_S \left[ \frac{1}{n^2} \sum_{i,j=1}^{n} \left( k(\mathbf{x}_i, \mathbf{x}_i) - \sum_{k=1}^{K} k(\mathbf{x}_i, \mathbf{x}_j) h_k^*(\mathbf{x}_i) h_k^*(\mathbf{x}_j) \right) \right]$$

(Because $\{\tilde{h}_k\}_{k=1}^{K}$ is the optimal solution on training dataset $S$.)

$$= \frac{1}{n^2} \sum_{i,j=1}^{n} \left[ \left( k(\mathbf{x}_i, \mathbf{x}_i) - \iint_{\mathbf{x}_i, \mathbf{x}_j} \sum_{k=1}^{K} k(\mathbf{x}_i, \mathbf{x}_j) h_k^*(\mathbf{x}_i) h_k^*(\mathbf{x}_j) d\mathbf{x}_i d\mathbf{x}_j \right) \right]$$

$$= \frac{1}{n^2} \sum_{i,j=1}^{n} \left[ \left( k(\mathbf{x}, \mathbf{x}) - \iint_{\mathbf{x}, \mathbf{y}} \sum_{k=1}^{K} k(\mathbf{x}, \mathbf{y}) h_k^*(\mathbf{x}) h_k^*(\mathbf{y}) d\mathbf{x} d\mathbf{y} \right) \right]$$

$$= R(H^*, \mathbb{P}).$$

The proof is complete.

$\square$

## A.3 Proof of Theorem 5.2

To prove Theorem 5.2, we need the following two lemmas. The first lemma is Lemma 3 in [24].

**Lemma A.1.** *[24] Let $\mathbf{A}$ be a $n \times n$ Hermitian matrix and let B be a Hermitian perturbation. Let $(\lambda, \mathbf{v})$ be the eigenvalue/vector pair of $\mathbf{A} + \mathbf{B}$ corresponding to $(\lambda_i, \mathbf{v}_i)$ of $\mathbf{A}$, let $\delta = \min_{j \neq i} |\lambda - \lambda_j|$, where $\{\lambda_j\}_{j=1}^n$ are all the eigenvalues of $\mathbf{A}$; then*

$$\sin \theta(\mathbf{v}, \mathbf{v}_i) \leq \frac{\|\mathbf{B}\|_{op}}{\delta},$$

*where $\theta(\mathbf{v}, \mathbf{v}_i)$ is defined as*

$$\theta(\mathbf{v}, \mathbf{v}_i) = \arccos \frac{\mathbf{v}^\top \mathbf{v}_i}{\|\mathbf{v}\| \|\mathbf{v}_i\|}.$$

**Lemma A.2.** *Let $\mathbf{A}$ be a $n \times n$ Hermitian matrix and let $\mathbf{B}$ be a $(n-1) \times (n-1)$ matrix which is constructed by deleting the $i$-th row and $i$-th column of $\mathbf{A}$. Assume that $\lambda_1 \geq \cdots \geq \lambda_n$ and $\mu_1 \geq \cdots \geq \mu_{n-1}$ are the eigenvalues of $\mathbf{A}$ and $\mathbf{B}$, respectively. Then,*

$$\lambda_1 \geq \mu_1 \geq \lambda_2 \geq \cdots \geq \lambda_{n-1} \geq \mu_{n-1} \geq \lambda_n$$

*always holds.*

*Proof.* Let $\mathbf{y}, \mathbf{z}_j \in \mathbb{R}^{n \times 1}$ be two vectors which are denoted as $\mathbf{y} = [y_1, \cdots, y_n]^\top$ and $\mathbf{z}_j = [z_{j1}, \cdots, z_{jn}]^\top$, respectively. We then define $\mathbf{x}, \mathbf{u}_i \in \mathbb{R}^{(n-1) \times 1}$ as follows:

$$\mathbf{x} = \underbrace{[y_1, \cdots, y_{i-1}, y_{i+1}, \cdots, y_n]^\top}_{\text{delete } i\text{-th component of } \mathbf{y}}, \qquad \mathbf{u}_j = \underbrace{[z_{j1}, \cdots, z_{j,i-1}, z_{j,i+1}, \cdots, z_{jn}]^\top}_{\text{delete } i\text{-th component of } \mathbf{z}_j}.$$

Then, by Courant-Fischer Minimax Theorem [7], we have

$$\lambda_j = \min_{\mathbf{z}_1, \cdots, \mathbf{z}_{j-1}} \max_{\substack{\mathbf{y} \perp \mathbf{z}_s \\ s=1, \cdots, j-1}} \frac{\mathbf{y}^\top \mathbf{A} \mathbf{y}}{\mathbf{y}^\top \mathbf{y}}$$

$$\geq \min_{\mathbf{z}_1, \cdots, \mathbf{z}_{j-1}} \max_{\substack{\mathbf{y} \perp \mathbf{z}_s \\ s=1, \cdots, j-1 \\ y_i=0}} \frac{\mathbf{y}^\top \mathbf{A} \mathbf{y}}{\mathbf{y}^\top \mathbf{y}}$$

$$= \min_{\mathbf{u}_1, \cdots, \mathbf{u}_{j-1}} \max_{\substack{\mathbf{x} \perp \mathbf{u}_s \\ s=1, \cdots, j-1}} \frac{\mathbf{x}^\top \mathbf{B} \mathbf{x}}{\mathbf{x}^\top \mathbf{x}}$$

$$= \mu_j.$$

Applying Rayleigh-Ritz Theorem [7], we have

$$\lambda_1 = \max_{\mathbf{y}} \frac{\mathbf{y}^\top \mathbf{A} \mathbf{y}}{\mathbf{y}^\top \mathbf{y}} \geq \max_{\substack{\mathbf{y} \\ y_i=0}} \frac{\mathbf{y}^\top \mathbf{A} \mathbf{y}}{\mathbf{y}^\top \mathbf{y}} = \max_{\mathbf{x}} \frac{\mathbf{x}^\top \mathbf{B} \mathbf{x}}{\mathbf{x}^\top \mathbf{x}} = \mu_1.$$

Moreover, by Courant-Fischer Minimax Theorem [7], we have

$$\lambda_{j+1} = \max_{\mathbf{z}_1, \cdots, \mathbf{z}_{n-j}} \min_{\substack{\mathbf{y} \perp \mathbf{z}_s \\ s=1, \cdots, n-j}} \frac{\mathbf{y}^\top \mathbf{A} \mathbf{y}}{\mathbf{y}^\top \mathbf{y}}$$

$$\leq \max_{\mathbf{z}_1, \cdots, \mathbf{z}_{n-j}} \min_{\substack{\mathbf{y} \perp \mathbf{z}_s \\ s=1, \cdots, n-j \\ y_i=0}} \frac{\mathbf{y}^\top \mathbf{A} \mathbf{y}}{\mathbf{y}^\top \mathbf{y}}$$

$$= \max_{\mathbf{u}_1, \cdots, \mathbf{u}_{n-j}} \min_{\substack{\mathbf{x} \perp \mathbf{u}_s \\ s=1, \cdots, n-j}} \frac{\mathbf{x}^\top \mathbf{B} \mathbf{x}}{\mathbf{x}^\top \mathbf{x}}$$

$$= \mu_j.$$

Applying Rayleigh-Ritz Theorem [7], we have

$$\lambda_n = \min_{\mathbf{y}} \frac{\mathbf{y}^\top \mathbf{A} \mathbf{y}}{\mathbf{y}^\top \mathbf{y}} \leq \min_{\substack{\mathbf{y} \\ y_i=0}} \frac{\mathbf{y}^\top \mathbf{A} \mathbf{y}}{\mathbf{y}^\top \mathbf{y}} = \min_{\mathbf{x}} \frac{\mathbf{x}^\top \mathbf{B} \mathbf{x}}{\mathbf{x}^\top \mathbf{x}} = \mu_{n-1}.$$

The proof is complete. $\square$

Now, we complete the proof of Theorem 5.2.

*Proof.* Notice that $\tilde{h}_k(\mathbf{x}) = \frac{1}{n\lambda_k}\phi(\mathbf{x})^\top \left(\sum_{j=1}^n h_{jk}\phi(\mathbf{x}_j)\right)$. Letting $\mathbf{a}_k = \sum_{j=1}^n h_{jk}\phi(\mathbf{x}_j)$, then we have

$$
\begin{aligned}
&|l(\mathbf{x}, \mathbf{y}, \tilde{H}) - l(\mathbf{x}, \mathbf{y}, \tilde{H}^i)| \\
&= \left|\sum_{k=1}^K \phi(\mathbf{x})^\top \phi(\mathbf{y})\tilde{h}_k(\mathbf{x})\tilde{h}_k(\mathbf{y}) - \sum_{k=1}^K \phi(\mathbf{x})^\top \phi(\mathbf{y})\tilde{h}_k^i(\mathbf{x})\tilde{h}_k^i(\mathbf{y})\right| \\
&= \left|\phi(\mathbf{x})^\top \phi(\mathbf{y})\right|\left|\sum_{k=1}^K \tilde{h}_k(\mathbf{x})\tilde{h}_k(\mathbf{y}) - \sum_{k=1}^K \tilde{h}_k^i(\mathbf{x})\tilde{h}_k^i(\mathbf{y})\right| \\
&\le \left|\sum_{k=1}^K \frac{1}{n^2\lambda_k^2}(\phi(\mathbf{x})^\top\mathbf{a}_k)(\phi(\mathbf{y})^\top\mathbf{a}_k) - \sum_{k=1}^K \frac{1}{(n\lambda_k^i)^2}(\phi(\mathbf{x})^\top\mathbf{a}_k^i)(\phi(\mathbf{y})^\top\mathbf{a}_k^i)\right| \\
&\qquad\left(\text{Because } \left|\phi(\mathbf{x})^\top\phi(\mathbf{y})\right| \le \frac{1}{2}\left(\|\phi(\mathbf{x})\|^2 + \|\phi(\mathbf{y})\|^2\right) \le 1\right). \\
&= \left|\sum_{k=1}^K \operatorname{Tr}\left(\left(\frac{1}{n^2\lambda_k^2}\mathbf{a}_k\mathbf{a}_k^\top - \frac{1}{(n\lambda_k^i)^2}\mathbf{a}_k^i\left(\mathbf{a}_k^i\right)^\top\right)\phi(\mathbf{y})\phi(\mathbf{x})^\top\right)\right| \\
&\le \sum_{k=1}^K \left|\operatorname{Tr}\left(\left(\frac{1}{n^2\lambda_k^2}\mathbf{a}_k\mathbf{a}_k^\top - \frac{1}{(n\lambda_k^i)^2}\mathbf{a}_k^i\left(\mathbf{a}_k^i\right)^\top\right)\phi(\mathbf{y})\phi(\mathbf{x})^\top\right)\right| \\
&\le \sum_{k=1}^K \sqrt{\operatorname{Tr}\left(\frac{1}{n^2\lambda_k^2}\mathbf{a}_k\mathbf{a}_k^\top - \frac{1}{(n\lambda_k^i)^2}\mathbf{a}_k^i\left(\mathbf{a}_k^i\right)^\top\right)^2}\sqrt{\operatorname{Tr}(\phi(\mathbf{y})\phi(\mathbf{x})^\top)^2} \\
&\le \sum_{k=1}^K \sqrt{\operatorname{Tr}\left(\frac{1}{n^2\lambda_k^2}\mathbf{a}_k\mathbf{a}_k^\top - \frac{1}{(n\lambda_k^i)^2}\mathbf{a}_k^i\left(\mathbf{a}_k^i\right)^\top\right)^2}. \\
&\qquad\left(\text{Because } \operatorname{Tr}\left(\phi(\mathbf{y})\phi(\mathbf{x})^\top\right)^2 = (\phi(\mathbf{x})^\top\phi(\mathbf{y}))^2 \le 1\right).
\end{aligned}
\tag{13}
$$

Denote that $\boldsymbol{\Phi} = [\phi(\mathbf{x}_1), ..., \phi(\mathbf{x}_n)]^\top \in \mathbb{R}^{n\times D}$, where $D$ is the dimension of feature space $\mathcal{H}$. Performing rank-$n$ singular value decomposition (SVD) on $\boldsymbol{\Phi}$, we have $\boldsymbol{\Phi} = \mathbf{H}\boldsymbol{\Sigma}\mathbf{V}^\top$, where $\mathbf{H} \in \mathbb{R}^{n\times n}$, $\boldsymbol{\Sigma} \in \mathbb{R}^{n\times n}$ is a diagonal matrix whose diagonal elements are the singular values of $\boldsymbol{\Phi}$, and $\mathbf{V} \in \mathbb{R}^{D\times n}$. Denote that $\mathbf{V} = [\mathbf{v}_1, ..., \mathbf{v}_n]$. We can obtain that $\mathbf{a}_k = \boldsymbol{\Phi}^\top\mathbf{h}_k = \mathbf{V}\boldsymbol{\Sigma}\mathbf{H}^\top\mathbf{h}_k$. Notice that

$$
\boldsymbol{\Sigma}\mathbf{H}^\top\mathbf{h}_k = \boldsymbol{\Sigma}\underbrace{[0, ..., 1, ..., 0]^\top}_{k\text{-th}} = \underbrace{[0, ..., \sigma_k, ..., 0]^\top}_{k\text{-th}}.
$$

Consequently, $\mathbf{a}_k = \sigma_k\mathbf{v}_k$, where $\sigma_k$ is the $k$-th singular value of $\boldsymbol{\Phi}$. By $\sigma_k^2 = n\lambda_k$, we have

$$
\begin{aligned}
&|l(\mathbf{x}, \mathbf{y}, H) - l(\mathbf{x}, \mathbf{y}, H^i)| \\
&\le \sum_{k=1}^K \sqrt{\operatorname{Tr}\left(\frac{1}{n\lambda_k}\mathbf{v}_k\mathbf{v}_k^\top - \frac{1}{n\lambda_k^i}\mathbf{v}_k^i\left(\mathbf{v}_k^i\right)^\top\right)^2}.
\end{aligned}
\tag{14}
$$

For the $k$-th item in Eq.(14), we have

$$\mathrm{Tr}\left(\frac{1}{n\lambda_k}\mathbf{v}_k\mathbf{v}_k^\top - \frac{1}{n\lambda_k^i}\mathbf{v}_k^i\left(\mathbf{v}_k^i\right)^\top\right)^2$$

$$=\mathrm{Tr}\left(\frac{1}{n\lambda_k}\mathbf{v}_k\mathbf{v}_k^\top - \frac{1}{n\lambda_k^i}\mathbf{v}_k\mathbf{v}_k^\top + \frac{1}{n\lambda_k^i}\mathbf{v}_k\mathbf{v}_k^\top - \frac{1}{n\lambda_k^i}\mathbf{v}_k^i\left(\mathbf{v}_k^i\right)^\top\right)^2$$

$$\leq 2\mathrm{Tr}\left(\frac{1}{n\lambda_k}\mathbf{v}_k\mathbf{v}_k^\top - \frac{1}{n\lambda_k^i}\mathbf{v}_k\mathbf{v}_k^\top\right)^2 + 2\mathrm{Tr}\left(\frac{1}{n\lambda_k^i}\mathbf{v}_k\mathbf{v}_k^\top - \frac{1}{n\lambda_k^i}\mathbf{v}_k^i\left(\mathbf{v}_k^i\right)^\top\right)^2 \quad (15)$$

$$=\underbrace{2\left(\frac{1}{n\lambda_k} - \frac{1}{n\lambda_k^i}\right)^2}_{\mathcal{A}} + \underbrace{\frac{2}{(n\lambda_k^i)^2}\mathrm{Tr}\left(\mathbf{v}_k\mathbf{v}_k^\top - \mathbf{v}_k^i\left(\mathbf{v}_k^i\right)^\top\right)^2}_{\mathcal{B}}.$$

We first bound $\mathcal{A}$ in Eq.(15). Let $\frac{1}{n}\mathbf{K}_n^{\backslash i}$ be the empirical kernel matrix whose $i$-th sample $\phi(\mathbf{x}_i) = \mathbf{0}$, and $\lambda_k^{\backslash i}$, $\mathbf{v}_k^{\backslash i}$ be its $k$-th eigenvalue/vector pair. Then, we have

$$(\lambda_k - \lambda_k^i)^2 = (\lambda_k - \lambda_k^{\backslash i} + \lambda_k^{\backslash i} - \lambda_k^i)^2 \leq 2(\lambda_k - \lambda_k^{\backslash i})^2 + 2(\lambda_k^{\backslash i} - \lambda_k^i)^2. \quad (16)$$

By Weyl inequality,

$$\lambda_k \leq \lambda_k^{\backslash i} + \frac{1}{n}\lambda_1\left(\boldsymbol{\Phi}\boldsymbol{\Phi}^\top - \boldsymbol{\Phi}^{\backslash i}\left(\boldsymbol{\Phi}^{\backslash i}\right)^\top\right).$$

Moreover,

$$\lambda_1\left(\boldsymbol{\Phi}\boldsymbol{\Phi}^\top - \boldsymbol{\Phi}^{\backslash i}\left(\boldsymbol{\Phi}^{\backslash i}\right)^\top\right)$$

$$=\lambda_1\left(\boldsymbol{\Phi}\boldsymbol{\Phi}^\top - \boldsymbol{\Phi}^{\backslash i}\boldsymbol{\Phi}^\top + \boldsymbol{\Phi}^{\backslash i}\boldsymbol{\Phi}^\top - \boldsymbol{\Phi}^{\backslash i}\left(\boldsymbol{\Phi}^{\backslash i}\right)^\top\right)$$

$$\leq\lambda_1\left(\boldsymbol{\Phi}\boldsymbol{\Phi}^\top - \boldsymbol{\Phi}^{\backslash i}\boldsymbol{\Phi}^\top\right) + \lambda_1\left(\boldsymbol{\Phi}^{\backslash i}\boldsymbol{\Phi}^\top - \boldsymbol{\Phi}^{\backslash i}\left(\boldsymbol{\Phi}^{\backslash i}\right)^\top\right)$$

$$\leq \max_{\|\mathbf{x}\|=1}\mathbf{x}^\top\left(\left(\boldsymbol{\Phi} - \boldsymbol{\Phi}^{\backslash i}\right)\boldsymbol{\Phi}^\top\right)\mathbf{x} + \max_{\|\mathbf{x}\|=1}\mathbf{x}^\top\left(\boldsymbol{\Phi}^{\backslash i}\left(\boldsymbol{\Phi} - \boldsymbol{\Phi}^{\backslash i}\right)^\top\right)\mathbf{x}$$

$$= \max_{\|\mathbf{x}\|=1}x_i\phi(\mathbf{x}_i)^\top\left(\sum_{j=1}^n x_j\phi(\mathbf{x}_j)\right) + \max_{\|\mathbf{x}\|=1}\left(\sum_{j\neq i}^n x_j\phi(\mathbf{x}_j)\right)^\top x_i\phi(\mathbf{x}_i) \quad (17)$$

$$\leq \max_{\|\mathbf{x}\|=1}|x_i|\left(\sum_{j=1}^n |x_j\phi^\top(\mathbf{x}_j)\phi(\mathbf{x}_i)|\right) + \max_{\|\mathbf{x}\|=1}|x_i|\left(\sum_{j\neq i}^n |x_j\phi^\top(\mathbf{x}_j)\phi(\mathbf{x}_i)|\right)$$

$$\leq 2\max_{\|\mathbf{x}\|=1}\sum_{j=1}^n |x_j|$$

$$\leq 2\max_{\|\mathbf{x}\|=1}\sqrt{n\left(\sum_{j=1}^n x_j^2\right)} = 2\sqrt{n}.$$

Thus, $\lambda_k \leq \lambda_k^{\backslash i} + \frac{2}{\sqrt{n}}$. On the other hand, denote $\frac{1}{n}\hat{\mathbf{K}}_n^{\backslash i}$ as a matrix constructed by deleting the $i$-th row and $i$-th column of $\frac{1}{n}\mathbf{K}_n$. Then, $\frac{1}{n}\hat{\mathbf{K}}_n^{\backslash i}$ and $\frac{1}{n}\mathbf{K}_n^{\backslash i}$ have the same non-zero eigenvalues. By Lemma A.2, we have $\lambda_k\left(\frac{1}{n}\mathbf{K}_n\right) \geq \lambda_k\left(\frac{1}{n}\hat{\mathbf{K}}_n^{\backslash i}\right) = \lambda_k\left(\frac{1}{n}\mathbf{K}_n^{\backslash i}\right)$, i.e., $\lambda_k \geq \lambda_k^{\backslash i}$. Thus, $2(\lambda_k - \lambda_k^{\backslash i})^2 \leq \frac{8}{n}$ holds. In the same way, we have $2(\lambda_k^{\backslash i} - \lambda_k^i)^2 \leq \frac{8}{n}$. Then, we can obtain

$$\mathcal{A} \leq \frac{2}{(n\lambda_k\lambda_k^i)^2}\left(\lambda_k - \lambda_k^i\right)^2 \leq \frac{4c^4}{n^2}((\lambda_k - \lambda_k^{\backslash i})^2 + (\lambda_k^{\backslash i} - \lambda_k^i)^2) \leq \frac{32c^4}{n^3}. \quad (18)$$

We then bound $\mathcal{B}$ in Eq.(15).

$$\mathcal{B} \leq \frac{2c^2}{n^2} \text{Tr} \left( \mathbf{v}_k \mathbf{v}_k^\top - \mathbf{v}_k^{\backslash i} \left( \mathbf{v}_k^{\backslash i} \right)^\top + \mathbf{v}_k^{\backslash i} \left( \mathbf{v}_k^{\backslash i} \right)^\top - \mathbf{v}_k^i \left( \mathbf{v}_k^i \right)^\top \right)^2$$

$$\leq \frac{4c^2}{n^2} \text{Tr} \left( \mathbf{v}_k \mathbf{v}_k^\top - \mathbf{v}_k^{\backslash i} \left( \mathbf{v}_k^{\backslash i} \right)^\top \right)^2 + \frac{4c^2}{n^2} \text{Tr} \left( \mathbf{v}_k^{\backslash i} \left( \mathbf{v}_k^{\backslash i} \right)^\top - \mathbf{v}_k^i \left( \mathbf{v}_k^i \right)^\top \right)^2 . \tag{19}$$

To bound the first item in Eq.(19), we need the help of Lemma A.1. We can deduce that

$$\text{Tr} \left( \mathbf{v}_k \mathbf{v}_k^\top - \mathbf{v}_k^{\backslash i} \left( \mathbf{v}_k^{\backslash i} \right)^\top \right)^2 = 2 - 2 \left( \mathbf{v}_k^\top \mathbf{v}_k^{\backslash i} \right)^2 = 2(1 - \cos^2 \theta(\mathbf{v}_k, \mathbf{v}_k^{\backslash i})) = 2 \sin^2 \theta(\mathbf{v}_k, \mathbf{v}_k^{\backslash i}).$$

To apply Lemma A.1, we must study the relationship between minimum eigenvalue gap of $\frac{1}{n}\mathbf{K}_n$ and $\delta$ in this lemma. By Lemma A.2, we have $\lambda_{k-1} \geq \lambda_k \geq \lambda_k^{\backslash i} \geq \lambda_{k+1}$. Thus, $\delta = \min\{\lambda_{k-1} - \lambda_k^{\backslash i}, \lambda_k^{\backslash i} - \lambda_{k+1}\}$. If $\delta = \lambda_{k-1} - \lambda_k^{\backslash i}$, we can obtain that $\delta \geq \lambda_{k-1} - \lambda_k \geq c_1$. If $\delta = \lambda_k^{\backslash i} - \lambda_{k+1}$, we can obtain that $\delta \geq \lambda_k - \lambda_{k+1} - \frac{2}{\sqrt{n}} \geq c_1 - \frac{2}{\sqrt{n}} \geq \frac{c_1}{2}$ because $\lambda_k \leq \lambda_k^{\backslash i} + \frac{2}{\sqrt{n}}$.
On the other hand, by the same derivation in Eq.(17) and $\mathbf{\Phi}^\top \mathbf{\Phi} \succcurlyeq \left( \mathbf{\Phi}^{\backslash i} \right)^\top \mathbf{\Phi}^{\backslash i}$, we can obtain

$$\left\| \frac{1}{n}\mathbf{\Phi}^\top \mathbf{\Phi} - \frac{1}{n} \left( \mathbf{\Phi}^{\backslash i} \right)^\top \mathbf{\Phi}^{\backslash i} \right\|_{op}^2 = \lambda_1 \left( \frac{1}{n}\mathbf{\Phi}^\top \mathbf{\Phi} - \frac{1}{n} \left( \mathbf{\Phi}^{\backslash i} \right)^\top \mathbf{\Phi}^{\backslash i} \right)^2 \leq \frac{2}{\sqrt{n}}. \text{ Above all, we have}$$

$$\text{Tr} \left( \mathbf{v}_k \mathbf{v}_k^\top - \mathbf{v}_k^{\backslash i} \left( \mathbf{v}_k^{\backslash i} \right)^\top \right)^2 \leq \frac{4 \left\| \frac{1}{n}\mathbf{\Phi}^\top \mathbf{\Phi} - \frac{1}{n} \left( \mathbf{\Phi}^{\backslash i} \right)^\top \mathbf{\Phi}^{\backslash i} \right\|_{op}^2}{c_1^2} \leq \frac{16}{c_1^2 n}.$$

The same bound of $\text{Tr} \left( \mathbf{v}_k^{\backslash i} \left( \mathbf{v}_k^{\backslash i} \right)^\top - \mathbf{v}_k^i \left( \mathbf{v}_k^i \right)^\top \right)^2$ can also be obtained by above process. Thus, we have

$$\mathcal{B} \leq \frac{128c^2}{c_1^2 n^3}. \tag{20}$$

Combining Eq.(18), Eq.(20) and Eq.(15), we know that there exist a constant $c_0$ such that

$$|l(\mathbf{x}, \mathbf{y}, \tilde{H}) - l(\mathbf{x}, \mathbf{y}, \tilde{H}^i)| \leq \frac{c_0 K}{n\sqrt{n}}.$$

holds. The proof is complete. $\qquad\square$

### A.4   Proof of Theorem 5.4

*Proof.* By the triangle inequality, we have

$$|l(\mathbf{x}, \mathbf{y}, \widetilde{H}_{\boldsymbol{\alpha}}) - l(\mathbf{x}, \mathbf{y}, \widetilde{H}_{\boldsymbol{\alpha}^i}^i)|$$
$$= |l(\mathbf{x}, \mathbf{y}, \widetilde{H}_{\boldsymbol{\alpha}}) - l(\mathbf{x}, \mathbf{y}, \widetilde{H}_{\boldsymbol{\alpha}^i}) + l(\mathbf{x}, \mathbf{y}, \widetilde{H}_{\boldsymbol{\alpha}^i}) - l(\mathbf{x}, \mathbf{y}, \widetilde{H}_{\boldsymbol{\alpha}^i}^i)|$$
$$\leq \underbrace{|l(\mathbf{x}, \mathbf{y}, \widetilde{H}_{\boldsymbol{\alpha}}) - l(\mathbf{x}, \mathbf{y}, \widetilde{H}_{\boldsymbol{\alpha}^i})|}_{\mathcal{A}} + \underbrace{|l(\mathbf{x}, \mathbf{y}, \widetilde{H}_{\boldsymbol{\alpha}^i}) - l(\mathbf{x}, \mathbf{y}, \widetilde{H}_{\boldsymbol{\alpha}^i}^i)|}_{\mathcal{B}}.$$

For $\mathcal{B}$, because $\mathbf{K}_{\boldsymbol{\alpha}} = \sum_{p=1}^m \alpha_p^2 \mathbf{K}_p$ is a kernel matrix and meets the conditions in Theorem 5.2, $\mathcal{B}$ can upper bounded by $\mathcal{O}\left( \frac{K}{n\sqrt{n}} \right)$.

Let $\boldsymbol{\alpha}^{(0)}, ..., \boldsymbol{\alpha}^{(m)} \in \mathbb{R}^m$ be the vectors which satisfy:

- $\boldsymbol{\alpha}^{(0)} = \boldsymbol{\alpha}$.

- $\boldsymbol{\alpha}^{(p-1)}$ and $\boldsymbol{\alpha}^{(p)}$ differ only by the $p$-th element ($p \in [m-1]$).

- The $p$-th element of $\boldsymbol{\alpha}^{(p)}$ is equal to the $p$-th element of $\boldsymbol{\alpha}^i$ ($p \in [m-1]$).

- $\boldsymbol{\alpha}^{(m)} = \boldsymbol{\alpha}^i$

By construction, $\mathcal{A}$ can be bounded as

$$\mathcal{A} = |l(\mathbf{x}, \mathbf{y}, \widetilde{H}_{\boldsymbol{\alpha}}) - l(\mathbf{x}, \mathbf{y}, \widetilde{H}_{\boldsymbol{\alpha}^i})|$$
$$\leq \sum_{p=1}^{m} \left| l\left(\mathbf{x}, \mathbf{y}, \widetilde{H}_{\boldsymbol{\alpha}^{(p-1)}}\right) - l\left(\mathbf{x}, \mathbf{y}, \widetilde{H}_{\boldsymbol{\alpha}^{(p)}}\right)\right|.$$

Denote the $k$-th eigenvalue/vector pair of $\frac{1}{n}\mathbf{K}_{\boldsymbol{\alpha}^{(p)}}$ and $\frac{1}{n}\mathbf{K}_{\boldsymbol{\alpha}^{(p-1)}}$ as $(\mu_k, \mathbf{v}_k)$ and $(\mu'_k, \mathbf{v}'_k)$, respectively. Then, by the proof of Theorem 5.2, we have

$$\left| l\left(\mathbf{x}, \mathbf{y}, \widetilde{H}_{\boldsymbol{\alpha}^{(p-1)}}\right) - l\left(\mathbf{x}, \mathbf{y}, \widetilde{H}_{\boldsymbol{\alpha}^{(p)}}\right)\right| \leq \sum_{k=1}^{K} \sqrt{\mathrm{Tr}\left(\frac{1}{n\mu_k}\mathbf{v}_k\mathbf{v}_k^\top - \frac{1}{n\mu'_k}\mathbf{v}'_k(\mathbf{v}'_k)^\top\right)^2}$$

$$\leq \sum_{k=1}^{K} \sqrt{\underbrace{2\left(\frac{1}{n\mu_k} - \frac{1}{n\mu'_k}\right)^2}_{\mathcal{C}} + \underbrace{\frac{2}{(n\mu'_k)^2}\mathrm{Tr}\left(\mathbf{v}_k\mathbf{v}_k^\top - \mathbf{v}'_k(\mathbf{v}'_k)^\top\right)^2}_{\mathcal{D}}}.$$

By Weyl's inequality, we can upper bound item $\mathcal{C}$ in the above equation as follows:

$$\mathcal{C} \leq \frac{(\mu_k - \mu'_k)^2}{(n\mu_k\mu'_k)^2} \leq \frac{c^4}{n^2}\lambda_1^2\left(\frac{1}{n}\left(\left(\alpha_p^{(p)}\right)^2 - \left(\alpha_p^{(p-1)}\right)^2\right)\mathbf{K}_p\right)$$

$$= \frac{c^4}{n^2}\left(\left(\alpha_p^{(p)}\right)^2 - \left(\alpha_p^{(p-1)}\right)^2\right)^2\lambda_1^2\left(\frac{1}{n}\mathbf{K}_p\right)$$

$$\leq \frac{4c^4}{n^2}\left(\alpha_p^{(p)} - \alpha_p^{(p-1)}\right)^2$$

$$\leq \frac{4c^4}{n^2}\eta^2.$$

On the other hand, denote that $\boldsymbol{\Phi}_{\boldsymbol{\alpha}} = [\alpha_1\boldsymbol{\Phi}_1, \cdots, \alpha_m\boldsymbol{\Phi}_m]$. Then, we can obtain that

$$\left\|\boldsymbol{\Phi}_{\boldsymbol{\alpha}^{(p-1)}}^\top\boldsymbol{\Phi}_{\boldsymbol{\alpha}^{(p-1)}} - \boldsymbol{\Phi}_{\boldsymbol{\alpha}^{(p)}}^\top\boldsymbol{\Phi}_{\boldsymbol{\alpha}^{(p)}}\right\|_{op}^2$$

$$= \left\|\boldsymbol{\Phi}_{\boldsymbol{\alpha}^{(p-1)}}^\top\boldsymbol{\Phi}_{\boldsymbol{\alpha}^{(p-1)}} - \boldsymbol{\Phi}_{\boldsymbol{\alpha}^{(p-1)}}^\top\boldsymbol{\Phi}_{\boldsymbol{\alpha}^{(p)}} + \boldsymbol{\Phi}_{\boldsymbol{\alpha}^{(p-1)}}^\top\boldsymbol{\Phi}_{\boldsymbol{\alpha}^{(p)}} - \boldsymbol{\Phi}_{\boldsymbol{\alpha}^{(p)}}^\top\boldsymbol{\Phi}_{\boldsymbol{\alpha}^{(p)}}\right\|_{op}^2$$

$$\leq 2\left\|\boldsymbol{\Phi}_{\boldsymbol{\alpha}^{(p-1)}}^\top\left(\boldsymbol{\Phi}_{\boldsymbol{\alpha}^{(p-1)}} - \boldsymbol{\Phi}_{\boldsymbol{\alpha}^{(p)}}\right)\right\|_{op}^2 + 2\left\|\left(\boldsymbol{\Phi}_{\boldsymbol{\alpha}^{(p-1)}} - \boldsymbol{\Phi}_{\boldsymbol{\alpha}^{(p)}}\right)^\top\boldsymbol{\Phi}_{\boldsymbol{\alpha}^{(p)}}\right\|_{op}^2$$

$$= 2\left\|\left(\boldsymbol{\Phi}_{\boldsymbol{\alpha}^{(p-1)}} - \boldsymbol{\Phi}_{\boldsymbol{\alpha}^{(p)}}\right)\boldsymbol{\Phi}_{\boldsymbol{\alpha}^{(p-1)}}^\top\right\|_{op}^2 + 2\left\|\boldsymbol{\Phi}_{\boldsymbol{\alpha}^{(p)}}\left(\boldsymbol{\Phi}_{\boldsymbol{\alpha}^{(p-1)}} - \boldsymbol{\Phi}_{\boldsymbol{\alpha}^{(p)}}\right)^\top\right\|_{op}^2$$

$$= 2\left(\alpha_p^{(p-1)} - \alpha_p^{(p)}\right)^2\left(\alpha_p^{(p-1)}\right)^2\|\mathbf{K}_p\|_{op}^2 + 2\left(\alpha_p^{(p-1)} - \alpha_p^{(p)}\right)^2\left(\alpha_p^{(p)}\right)^2\|\mathbf{K}_p\|_{op}^2$$

$$\leq 4n^2\eta^2.$$

By the same technique in the proof of Theorem 5.2, we can upper bound item $\mathcal{D}$ as follows:

$$\mathcal{D} \leq \frac{2c^2}{n^2}\mathrm{Tr}\left(\mathbf{v}_k\mathbf{v}_k^\top - \mathbf{v}'_k(\mathbf{v}'_k)^\top\right)^2$$

$$\leq \frac{2c^2}{n^2} \cdot \frac{4\left\|\boldsymbol{\Phi}_{\boldsymbol{\alpha}^{(p-1)}}^\top\boldsymbol{\Phi}_{\boldsymbol{\alpha}^{(p-1)}}/n - \boldsymbol{\Phi}_{\boldsymbol{\alpha}^{(p)}}^\top\boldsymbol{\Phi}_{\boldsymbol{\alpha}^{(p)}}/n\right\|_{op}^2}{c_1^2}$$

$$\leq \frac{32c^2}{c_1^2 n^2}\eta^2.$$

Thus, we can obtain that $\left| l\left(\mathbf{x}, \mathbf{y}, \widetilde{H}_{\boldsymbol{\alpha}^{(p-1)}}\right) - l\left(\mathbf{x}, \mathbf{y}, \widetilde{H}_{\boldsymbol{\alpha}^{(p)}}\right) \right|$ has an upper bound as $\mathcal{O}\left(\frac{K\eta}{n}\right)$.

Above all, we have that there exists a constant $c_0$ such that

$$|l(\mathbf{x}, \mathbf{y}, \widetilde{H}_{\boldsymbol{\alpha}}) - l(\mathbf{x}, \mathbf{y}, \widetilde{H}_{\boldsymbol{\alpha}^i}^i)| \leq \frac{c_0 mK}{n}\eta + \frac{c_0 K}{n\sqrt{n}}.$$

$\square$

## A.5 The Optimization of SimpleMKKM

SimpleMKKM aims to solve the following kernel alignment-based optimization problem:

$$\min_{\boldsymbol{\alpha} \in \triangle} F(\boldsymbol{\alpha}) \tag{21}$$

where $F(\boldsymbol{\alpha}) = \max_{\mathbf{H}} \frac{1}{n} \mathrm{Tr}\left(\mathbf{K}_{\boldsymbol{\alpha}} \mathbf{H} \mathbf{H}^\top\right)$, s.t. $\mathbf{H}^\top \mathbf{H} = \mathbf{I}_K$, and $\triangle = \{\boldsymbol{\alpha} \in \mathbb{R}^m | \sum_{p=1}^m \alpha_p = 1, \alpha_p \geq 0, \forall p \in [m]\}$.

$F(\boldsymbol{\alpha})$ in Eq.(21) is proven differentiable and the $p$-th component of the gradient is $\frac{\partial F(\boldsymbol{\alpha})}{\partial \alpha_p} = \frac{2\alpha_p}{n}\mathrm{Tr}\left(\mathbf{K}_p \mathbf{H}^* \mathbf{H}^{*\top}\right)$, where $\mathbf{H}^* = \mathrm{argmax}_{\mathbf{H}^\top \mathbf{H} = \mathbf{I}_K} \frac{1}{n}\mathrm{Tr}\left(\mathbf{K}_{\boldsymbol{\alpha}} \mathbf{H} \mathbf{H}^\top\right)$. Then, a reduced gradient descent algorithm [26] is adopted to optimize Eq.(21). To keep the reduced gradient in an appropriate size, we scale it as

$$[\nabla F(\boldsymbol{\alpha})]_p = \frac{1}{nmK}\left(\frac{\partial F(\boldsymbol{\alpha})}{\partial \alpha_p} - \frac{\partial F(\boldsymbol{\alpha})}{\partial \alpha_u}\right)$$

and

$$[\nabla F(\boldsymbol{\alpha})]_u = \frac{1}{nmK} \sum_{p=1, p \neq u}^m \left(\frac{\partial F(\boldsymbol{\alpha})}{\partial \alpha_u} - \frac{\partial F(\boldsymbol{\alpha})}{\partial \alpha_p}\right).$$

where $u$ is the index of the largest element of $\boldsymbol{\alpha}$. In fact, the reduced gradient adopted in this paper is the original one [20] divided by $mK$, and it can also make SimpleMKKM converge within several iterations by the rescaled reduced gradient. Then, the descent direction $\mathbf{d} = [d_1, \cdots, d_m]^\top$ for optimizing the kernel weights $\boldsymbol{\alpha}$ can be computed as

$$d_p = \begin{cases} 0, & if \ \alpha_p = 0 \ \& \ [\nabla F(\boldsymbol{\alpha})]_p > 0, \\ -[\nabla F(\boldsymbol{\alpha})]_p, & if \ \alpha_p > 0 \ \& \ p \neq u, \\ -[\nabla F(\boldsymbol{\alpha})]_u, & if \ p = u. \end{cases}$$

The updating scheme of $\boldsymbol{\alpha}$ is $\boldsymbol{\alpha} \leftarrow \boldsymbol{\alpha} + \xi \mathbf{d}$, where $\xi$ is the learning step size. The optimal $\xi$ can be selected by Armijo's rule.

## A.6 Proof of Theorem 5.5

*Proof.* Denote the kernel weights obtained by performing SimpleMKKM on $S$ and $S^i$ as $\boldsymbol{\alpha}$ and $\boldsymbol{\beta}$, respectively. After $t$ iterations, denote that $\boldsymbol{\alpha}^{(t)}$ and $\boldsymbol{\beta}^{(t)}$ are the corresponding kernel weights, and $\mathbf{H}_{(t)}, \mathbf{H}_{(t)}^i$ are the clustering indicator matrices obtained by $\frac{1}{n}\mathbf{K}_{\boldsymbol{\alpha}^{(t)}}, \frac{1}{n}\mathbf{K}_{\boldsymbol{\beta}^{(t)}}^i$, respectively. Denote the maximal learning step as $\xi$.

Then, for any $p \in [m]$, we have

$$|\alpha_p^{(t+1)} - \beta_p^{(t+1)}| - |\alpha_p^{(t)} - \beta_p^{(t)}|$$

$$\leq 2\xi(m-1) \max_{p \in [m]} \left| \frac{1}{nmK}\alpha_p^{(t)} tr(\mathbf{H}_{(t)}^\top \mathbf{K}_p \mathbf{H}_{(t)}) - \frac{1}{nmK}\beta_p^{(t)} tr((\mathbf{H}_{(t)}^i)^\top \mathbf{K}_p^i \mathbf{H}_{(t)}^i) \right|$$

$$\leq 2\xi \max_{p \in [m]} \left| \frac{2}{nK}\alpha_p^{(t)} tr(\mathbf{H}_{(t)}^\top \mathbf{K}_p \mathbf{H}_{(t)}) - \frac{2}{nK}\beta_p^{(t)} tr(\mathbf{H}_{(t)}^\top \mathbf{K}_p \mathbf{H}_{(t)}) \right| \tag{1}$$

$$+ 2\xi \max_{p \in [m]} \left| \frac{2}{nK}\beta_p^{(t)} tr(\mathbf{H}_{(t)}^\top \mathbf{K}_p \mathbf{H}_{(t)}) - \frac{2}{nK}\beta_p^{(t)} tr((\mathbf{H}_{(t)}^i)^\top \mathbf{K}_p \mathbf{H}_{(t)}^i)) \right| \tag{2}$$

$$+ 2\xi \max_{p \in [m]} \left| \frac{2}{nK}\beta_p^{(t)} tr((\mathbf{H}_{(t)}^i)^\top \mathbf{K}_p \mathbf{H}_{(t)}^i)) - \frac{2}{nK}\beta_p^{(t)} tr((\mathbf{H}_{(t)}^i)^\top \mathbf{K}_p^i \mathbf{H}_{(t)}^i)) \right|. \tag{3}$$

For Item $(1)$, we have

$$(1) = 2\xi \max_{p\in[m]} |\alpha_p^{(t)} - \beta_p^{(t)}| \cdot \frac{tr(\mathbf{H}_{(t)}^\top \mathbf{K}_p \mathbf{H}_{(t)})}{nK} \le 2\xi\|\boldsymbol{\alpha}^{(t)} - \boldsymbol{\beta}^{(t)}\|_\infty.$$

For Item $(2)$, let $\boldsymbol{\Pi}_{\boldsymbol{\alpha}^{(t)}} = \mathbf{H}_{(t)}\mathbf{H}_{(t)}^\top$ and $\boldsymbol{\Pi}_{\boldsymbol{\beta}^{(t)}}^i = \mathbf{H}_{(t)}^i(\mathbf{H}_{(t)}^i)^\top$. We can obtain

$$
\begin{aligned}
(2) =& 2\xi \max_{p\in[m]} \beta_p^{(t)} \cdot tr\left(\frac{\mathbf{K}_p}{nK}(\boldsymbol{\Pi}_{\boldsymbol{\alpha}^{(t)}} - \boldsymbol{\Pi}_{\boldsymbol{\beta}^{(t)}}^i)\right)\\
\le& \frac{2\xi}{K} \max_{p\in[m]} \left\|\frac{\mathbf{K}_p}{n}\right\|_F \cdot \left\|\boldsymbol{\Pi}_{\boldsymbol{\alpha}^{(t)}} - \boldsymbol{\Pi}_{\boldsymbol{\beta}^{(t)}}^i\right\|_F\\
\le& \frac{2\xi}{K} \max_{p\in[m]} \sqrt{\frac{tr(\mathbf{K}_p^2)}{n^2}} \cdot \left\|\boldsymbol{\Pi}_{\boldsymbol{\alpha}^{(t)}} - \boldsymbol{\Pi}_{\boldsymbol{\beta}^{(t)}}^i\right\|_F\\
\le& \frac{2\xi}{K} \max_{p\in[m]} \sqrt{\frac{tr^2(\mathbf{K}_p)}{n^2}} \left\|\boldsymbol{\Pi}_{\boldsymbol{\alpha}^{(t)}} - \boldsymbol{\Pi}_{\boldsymbol{\beta}^{(t)}} + \boldsymbol{\Pi}_{\boldsymbol{\beta}^{(t)}} - \boldsymbol{\Pi}_{\boldsymbol{\beta}^{(t)}}^i\right\|_F\\
\le& \frac{2\xi}{K} \left\|\boldsymbol{\Pi}_{\boldsymbol{\alpha}^{(t)}} - \boldsymbol{\Pi}_{\boldsymbol{\beta}^{(t)}}\right\|_F + \frac{2\xi}{K} \left\|\boldsymbol{\Pi}_{\boldsymbol{\beta}^{(t)}} - \boldsymbol{\Pi}_{\boldsymbol{\beta}^{(t)}}^i\right\|_F.
\end{aligned}
$$

We can decompose $\boldsymbol{\Pi}_{\boldsymbol{\alpha}^{(t)}}$ and $\boldsymbol{\Pi}_{\boldsymbol{\beta}^{(t)}}$ as $\boldsymbol{\Pi}_{\boldsymbol{\alpha}^{(t)}} = \sum_{k=1}^K \mathbf{h}_k\mathbf{h}_k^\top$ and $\boldsymbol{\Pi}_{\boldsymbol{\beta}^{(t)}} = \sum_{k=1}^K \mathbf{u}_k\mathbf{u}_k^\top$, where $\mathbf{h}_k$ and $\mathbf{u}_k$ are the $k$-th eigenvectors of $\frac{1}{n}\mathbf{K}_{\boldsymbol{\alpha}^{(t)}}$ and $\frac{1}{n}\mathbf{K}_{\boldsymbol{\beta}^{(t)}}$, respectively. Then, using Lemma A.1 and the same deduction of Theorem 5.2, we have

$$
\begin{aligned}
\left\|\boldsymbol{\Pi}_{\boldsymbol{\alpha}^{(t)}} - \boldsymbol{\Pi}_{\boldsymbol{\beta}^{(t)}}\right\|_F =& \left\|\sum_{k=1}^K \mathbf{h}_k\mathbf{h}_k^\top - \sum_{k=1}^K \mathbf{u}_k\mathbf{u}_k^\top\right\|_F\\
\le& \sum_{k=1}^K \left\|\mathbf{h}_k\mathbf{h}_k^\top - \mathbf{u}_k\mathbf{u}_k^\top\right\|_F\\
=& \sum_{k=1}^K \sqrt{\mathrm{Tr}\left(\mathbf{h}_k\mathbf{h}_k^\top - \mathbf{u}_k\mathbf{u}_k^\top\right)^2}\\
\le& \frac{2K\left\|\frac{1}{n}\mathbf{K}_{\boldsymbol{\alpha}^{(t)}} - \frac{1}{n}\mathbf{K}_{\boldsymbol{\beta}^{(t)}}\right\|_{op}}{c_1}\\
=& \frac{2K\left\|\frac{1}{n}\sum_{p=1}^m \left(\left(\alpha_p^{(t)}\right)^2 - \left(\beta_p^{(t)}\right)^2\right)\mathbf{K}_p\right\|_{op}}{c_1}\\
\le& \frac{2K\sum_{p=1}^m \left|\left(\alpha_p^{(t)}\right)^2 - \left(\beta_p^{(t)}\right)^2\right|\left\|\frac{1}{n}\mathbf{K}_p\right\|_{op}}{c_1}\\
\le& \frac{2K\sum_{p=1}^m \left|\alpha_p^{(t)} - \beta_p^{(t)}\right|\left|\alpha_p^{(t)} + \beta_p^{(t)}\right|}{c_1}\\
\le& \|\boldsymbol{\alpha}^{(t)} - \boldsymbol{\beta}^{(t)}\|_\infty \frac{2K\sum_{p=1}^m \left|\alpha_p^{(t)} + \beta_p^{(t)}\right|}{c_1}\\
=& \frac{4K}{c_1}\|\boldsymbol{\alpha}^{(t)} - \boldsymbol{\beta}^{(t)}\|_\infty.
\end{aligned}
$$

Similar, we can obtain that

$$\left\|\boldsymbol{\Pi}_{\boldsymbol{\beta}^{(t)}} - \boldsymbol{\Pi}_{\boldsymbol{\beta}^{(t)}}^i\right\|_F \le \frac{2K\left\|\frac{1}{n}\mathbf{K}_{\boldsymbol{\beta}^{(t)}} - \frac{1}{n}\mathbf{K}_{\boldsymbol{\beta}^{(t)}}^i\right\|_{op}}{c_1} \le \frac{4K}{c_1\sqrt{n}}.$$

Thus, Item $(2)$ can be bounded as

$$(2) \le \frac{8\xi}{c_1}\|\boldsymbol{\alpha}^{(t)} - \boldsymbol{\beta}^{(t)}\|_\infty + \frac{8\xi}{c_1\sqrt{n}}.$$

For Item (3), we have

$$
\begin{aligned}
(3) &= \max_{p \in [m]} \frac{2\beta_p^{(t)} \xi}{nK} \mathrm{Tr}((\mathbf{K}_p - \mathbf{K}_p^i)\mathbf{H}_{(t)}^i (\mathbf{H}_{(t)}^i)^\top) \\
&\leq \frac{2\xi}{nK} \max_{p \in [m]} \left\| \mathbf{K}_p - \mathbf{K}_p^i \right\|_F \left\| \mathbf{H}_{(t)}^i (\mathbf{H}_{(t)}^i)^\top \right\|_F \\
&= \frac{2\xi}{n} \max_{p \in [m]} \left\| \mathbf{K}_p - \mathbf{K}_p^{\backslash i} + \mathbf{K}_p^{\backslash i} - \mathbf{K}_p^i \right\|_F \\
&\leq \frac{2\xi}{n} \max_{p \in [m]} \left\| \mathbf{K}_p - \mathbf{K}_p^{\backslash i} \right\|_F + \frac{2\xi}{n} \max_{p \in [m]} \left\| \mathbf{K}_p^{\backslash i} - \mathbf{K}_p^i \right\|_F \\
&= \frac{2\xi}{n} \max_{p \in [m]} \sqrt{\sum_{j=1}^n k_p^2(\mathbf{x}_i, \mathbf{x}_j) + \sum_{j \neq i}^n k_p^2(\mathbf{x}_i, \mathbf{x}_j)} + \frac{2}{n} \max_{p \in [m]} \sqrt{2\sum_{j \neq i}^n k_p^2(\mathbf{x}_i', \mathbf{x}_j) + k_p^2(\mathbf{x}_i', \mathbf{x}_i')} \\
&\leq \frac{2\xi}{n} \max_{p \in [m]} \sqrt{2\sum_{j=1}^n |k_p(\mathbf{x}_i, \mathbf{x}_j)|} + \frac{2\xi}{n} \max_{p \in [m]} \sqrt{2\sum_{j \neq i}^n |k_p(\mathbf{x}_i', \mathbf{x}_j)| + |k_p(\mathbf{x}_i', \mathbf{x}_i')|} \\
&\leq \frac{4\sqrt{2}\xi}{\sqrt{n}}.
\end{aligned}
$$

Combining the upper bounds of Item (1), (2) and (3), we can obtain that there exists a constant $c_0 \geq 1$ such that

$$
\|\boldsymbol{\alpha}^{(t+1)} - \boldsymbol{\beta}^{(t+1)}\|_\infty \leq c_0 \|\boldsymbol{\alpha}^{(t)} - \boldsymbol{\beta}^{(t)}\|_\infty + \frac{c_0}{\sqrt{n}}.
$$

Assume that the number of iterations is $T$. Then, by the same initialization, we know that

$$
\begin{aligned}
\|\boldsymbol{\alpha}^{(T)} - \boldsymbol{\beta}^{(T)}\|_\infty &\leq c_0 \|\boldsymbol{\alpha}^{(T-1)} - \boldsymbol{\beta}^{(T-1)}\|_\infty + \frac{c_0}{\sqrt{n}} \\
&\leq c_0^2 \|\boldsymbol{\alpha}^{(T-2)} - \boldsymbol{\beta}^{(T-2)}\|_\infty + (1 + c_0)\frac{c_0}{\sqrt{n}} \\
&\leq \cdots \\
&\leq c_0^T \|\boldsymbol{\alpha}^{(0)} - \boldsymbol{\beta}^{(0)}\|_\infty + (1 + c_0 + \cdots + c_0^{T-1})\frac{c_0}{\sqrt{n}} \\
&\leq \frac{T c_0^T}{\sqrt{n}}.
\end{aligned}
$$

The proof is complete.

$\square$

# B Further Experimental Results

## B.1 The Details of Datasets

The detailed information of five benchmark datasets, including *Flo17*[4], *Flo102*[5], *DIGIT*[6], *Cal102*[7] and *Reuters*[8], are listed in Table 3. The information of three large-scale datasets, including *NUSWIDE*[9], *AwA*[10] and *MNIST*[11] and *YtVideo*[12], are reported by Table 4. Among these datasets, NUSWIDE is an object image dataset that has 30000 samples and 31 classes. AwA is about the attributes of animals, which consists of 30475 images of 50 animal classes depicted in 6 views. To construct multiple views of MNIST, we adopt VGG19 [28], DenseNet121 [11], and ResNet101 [10] as the feature extractors for three views, respectively. The three deep neural networks are pre-trained on the ImageNet[5]. YtVideo is a dataset that consists of 101499 videos from Youtube.

Table 3: Benchmark datasets

| Datasets | Samples | Kernels | Clusters |
|----------|---------|---------|----------|
| Flo17 | 1360 | 7 | 17 |
| DIGIT | 2000 | 3 | 10 |
| CCV | 6773 | 3 | 20 |
| Flo102 | 8189 | 4 | 102 |
| Reuters | 18758 | 5 | 6 |

Table 4: Large-scale datasets used in the experiments

| Dataset | Samples | View | Clusters |
|---------|---------|------|----------|
| NUSWIDE | 30000 | 5 | 31 |
| AwA | 30475 | 6 | 50 |
| MNIST | 60000 | 3 | 10 |
| YtVideo | 101499 | 5 | 31 |

## B.2 Clustering Performance with Different Numbers of Landmarks

To study the clustering performance of the proposed method with different numbers of landmarks, we conduct relevant experiments on CCV and Flo102. Specifically, we vary the number of landmarks in $[200, \cdots, 2000]$ and record the corresponding ACC in Figure 1. The magenta curve denotes the variation of the proposed method. As a reference, we use the brown curve to illustrate the ACC of the original SimpleMKKM. As seen, as the number of landmarks increases, the ACC of the proposed method is approaching SimpleMKKM, and tends to be stable. Notice that the sample numbers of CCV and Flo102 are 6773 and 8189, respectively. It shows that we don't need too many landmarks for the comparable clustering performance of the original SimpleMKKM.

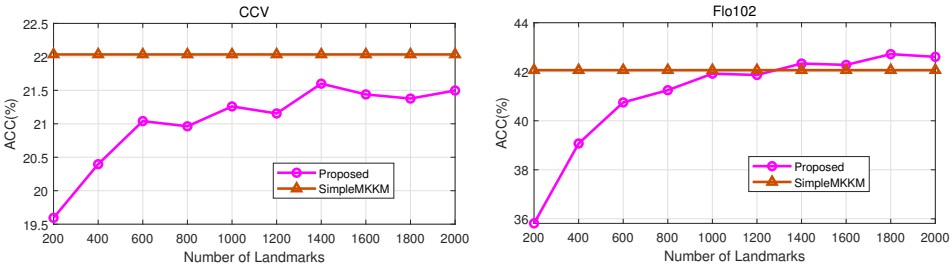

Figure 1: ACC comparison with different numbers of landmarks.

---

[4] www.robots.ox.ac.uk/~vgg/data/flowers/17/

[5] www.robots.ox.ac.uk/~vgg/data/flowers/102/

[6] http://ss.sysu.edu.cn/py/

[7] www.vision.caltech.edu/Image_Datasets/Caltech101/

[8] http://kdd.ics.uci.edu/databases/reuters21578/

[9] http://lms.comp.nus.edu.sg/wp-content/uploads/2019/research/nuswide/NUS-WIDE.html

[10] http://cvml.ist.ac.at/AwA/

[11] http://yann.lecun.com/exdb/mnist/

[12] http://archive.ics.uci.edu/ml/datasets/YouTube+Multiview+Video+Games+Dataset

### B.3 Empirical Analysis of Convergence

In this paper, we change the reduced gradients used in [20]. To study the optimization effect with new reduced gradients defined in Section A.5, we record the variation of the objective function after each iteration in Figure 2. As illustrated, the algorithm still converges within 5 iterations. Moreover, it also indicates that the number of iterations $T$ can be regarded as a constant.

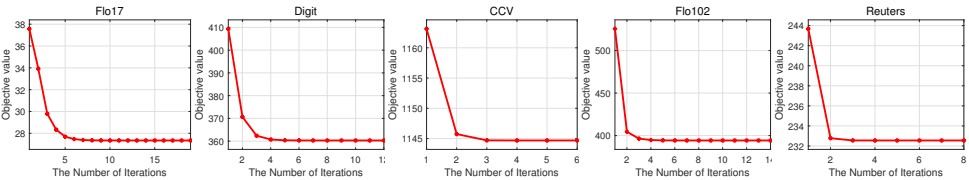

Figure 2: Convergence of SimpleMKKM

### B.4 Empirical Study of the Eigenvalues of Kernel Matrix

To verify the assumptions about the eigenvalues of the empirical kernel matrix in Theorem 5.2, we conduct experiments on the first view of MNIST with different kernel functions. Table 5 and Table 6 report the largest 11 eigenvalues of the kernel matrices constructed by Gaussian and linear kernel with different sample numbers, respectively. The sample numbers vary in $[1000, \cdots, 15000]$. As seen, the eigenvalues and their gap are stable with the variation of the sample number. This demonstrates that our assumptions are rational.

Table 5: The eigenvalues of kernel matrix constructed by Gaussian kernel function $k(\mathbf{x}, \mathbf{y}) = \exp(-\|\mathbf{x} - \mathbf{y}\|^2/\sigma^2)$ with different sample number. The used dataset is uniformly sampled from the first view of MNIST.

| Sample Number | $\lambda_1$ | $\lambda_2$ | $\lambda_3$ | $\lambda_4$ | $\lambda_5$ | $\lambda_6$ | $\lambda_7$ | $\lambda_8$ | $\lambda_9$ | $\lambda_{10}$ | $\lambda_{11}$ |
|---|---|---|---|---|---|---|---|---|---|---|---|
| | | | | | | $\times 10^{-3}$ | | | | | |
| 1000 | 383.38 | 60.41 | 52.85 | 44.62 | 36.63 | 31.01 | 29.55 | 25.77 | 22.34 | 20.95 | 5.28 |
| 2000 | 383.01 | 60.34 | 54.19 | 45.25 | 34.87 | 31.30 | 27.90 | 25.42 | 21.82 | 20.12 | 5.24 |
| 3000 | 382.87 | 60.42 | 53.38 | 42.48 | 35.64 | 30.88 | 29.16 | 23.10 | 22.53 | 20.94 | 5.27 |
| 4000 | 383.05 | 61.30 | 52.41 | 43.80 | 35.85 | 32.04 | 27.66 | 24.04 | 22.58 | 21.30 | 5.26 |
| 5000 | 382.67 | 58.24 | 52.04 | 44.88 | 35.06 | 31.24 | 28.69 | 24.30 | 22.51 | 20.75 | 5.25 |
| 6000 | 382.86 | 60.30 | 52.34 | 43.64 | 36.09 | 31.26 | 28.75 | 24.14 | 22.36 | 21.01 | 5.35 |
| 7000 | 382.91 | 59.82 | 53.51 | 43.47 | 35.15 | 31.65 | 28.17 | 24.38 | 22.35 | 21.38 | 5.26 |
| 8000 | 382.88 | 59.91 | 53.49 | 43.21 | 35.61 | 31.23 | 28.94 | 23.77 | 22.57 | 21.28 | 5.29 |
| 9000 | 382.84 | 60.09 | 52.24 | 44.60 | 35.40 | 31.37 | 28.55 | 24.00 | 22.31 | 20.92 | 5.22 |
| 10000 | 382.86 | 59.52 | 53.52 | 44.05 | 35.16 | 31.39 | 28.47 | 24.09 | 22.59 | 21.50 | 5.26 |
| 11000 | 382.82 | 59.53 | 52.91 | 43.74 | 35.68 | 31.56 | 28.57 | 24.03 | 22.43 | 21.08 | 5.29 |
| 12000 | 382.85 | 60.15 | 53.29 | 44.00 | 35.72 | 31.21 | 28.54 | 24.15 | 22.39 | 20.97 | 5.21 |
| 13000 | 382.79 | 59.69 | 53.11 | 44.16 | 35.39 | 31.12 | 28.62 | 24.06 | 22.32 | 21.04 | 5.23 |
| 14000 | 382.81 | 59.98 | 52.85 | 44.13 | 35.48 | 31.25 | 28.53 | 24.06 | 22.35 | 21.08 | 5.27 |
| 15000 | 382.82 | 60.01 | 53.12 | 44.01 | 35.40 | 31.19 | 28.55 | 24.05 | 22.37 | 21.10 | 5.24 |

### B.5 Experiments on Other Multiple Kernel Clustering Algorithms

To give more empirical studies of the proposed method, we conduct additional experiments on three classic algorithms, i.e., average multiple kernel $k$-means (AMKKM), multiple kernel $k$-means (MKKM) [12] and multiple kernel k-means clustering with matrix-induced regularization (MKKMMR) [21]. The results are reported in the following three tables. As seen from Table 7 and Table 8, our method achieves comparable clustering performance in comparison with the standard AMKKM and MKKM, while the running time is far less. However, as shown in Table 9, the results of our method fluctuate more dramatically when we apply it to MKKMMR. Two main reasons cause it: 1) The hyper-parameter of MKKMMR makes the kernel weights unstable; 2) The optimal hyper-parameter of MKKMMR on landmarks is different from the whole training dataset. Through the experimental results and our empirical analysis, our method would be more effective in the parameter-free multiple kernel clustering algorithms.

Table 6: The eigenvalues of kernel matrix constructed by linear kernel function $k(\mathbf{x}, \mathbf{y}) = \mathbf{x}^\top \mathbf{y}$ with different sample number. The used dataset is uniformly sampled from the first view of MNIST.

| Sample Number | $\lambda_1$ | $\lambda_2$ | $\lambda_3$ | $\lambda_4$ | $\lambda_5$ | $\lambda_6$ | $\lambda_7$ | $\lambda_8$ | $\lambda_9$ | $\lambda_{10}$ | $\lambda_{11}$ |
|---|---|---|---|---|---|---|---|---|---|---|---|
| | | | | | | $\times 10^{-3}$ | | | | | |
| 1000 | 721.76 | 69.18 | 56.74 | 39.65 | 27.80 | 23.41 | 16.92 | 11.52 | 10.04 | 7.20 | 2.27 |
| 2000 | 721.97 | 65.85 | 59.94 | 39.72 | 27.48 | 22.36 | 17.36 | 11.49 | 10.12 | 7.55 | 2.27 |
| 3000 | 722.41 | 65.88 | 59.47 | 39.24 | 26.94 | 22.97 | 17.47 | 11.33 | 10.27 | 7.63 | 2.47 |
| 4000 | 721.16 | 67.05 | 58.54 | 40.65 | 26.59 | 23.29 | 17.13 | 11.70 | 10.27 | 7.27 | 2.41 |
| 5000 | 720.70 | 67.71 | 60.08 | 39.82 | 26.57 | 22.54 | 17.03 | 11.29 | 10.36 | 7.56 | 2.40 |
| 6000 | 720.04 | 66.73 | 59.43 | 41.08 | 27.19 | 22.76 | 17.16 | 11.57 | 10.13 | 7.68 | 2.40 |
| 7000 | 720.83 | 66.69 | 59.57 | 40.26 | 27.48 | 22.64 | 17.33 | 11.19 | 10.18 | 7.56 | 2.37 |
| 8000 | 720.66 | 66.97 | 59.15 | 40.10 | 27.46 | 22.80 | 17.55 | 11.23 | 10.14 | 7.66 | 2.36 |
| 9000 | 720.63 | 66.56 | 59.69 | 40.50 | 27.20 | 22.44 | 17.42 | 11.39 | 10.27 | 7.60 | 2.39 |
| 10000 | 721.12 | 66.09 | 58.60 | 40.93 | 27.52 | 22.77 | 17.55 | 11.51 | 10.22 | 7.47 | 2.36 |
| 11000 | 721.43 | 66.22 | 59.10 | 40.41 | 27.09 | 22.75 | 17.57 | 11.31 | 10.25 | 7.52 | 2.39 |
| 12000 | 720.87 | 65.94 | 58.85 | 41.01 | 27.45 | 22.96 | 17.43 | 11.43 | 10.25 | 7.51 | 2.39 |
| 13000 | 720.73 | 66.41 | 59.16 | 40.45 | 27.32 | 22.92 | 17.41 | 11.41 | 10.26 | 7.62 | 2.40 |
| 14000 | 720.98 | 66.59 | 58.71 | 40.55 | 27.50 | 22.75 | 17.40 | 11.38 | 10.24 | 7.60 | 2.38 |
| 15000 | 720.99 | 66.23 | 59.05 | 40.53 | 27.39 | 22.84 | 17.44 | 11.40 | 10.24 | 7.58 | 2.38 |

Table 7: Experimental results of the proposed method in comparison with AMKKM.

| | Flo17 | | Digit | | CCV | | Flo102 | | Reuters | |
|---|---|---|---|---|---|---|---|---|---|---|
| | AMKKM | Approx. | AMKKM | Approx. | AMKKM | Approx. | AMKKM | Approx. | AMKKM | Approx. |
| ACC | 51.03 | 54.41 | 88.75 | 90.70 | 19.74 | 19.70 | 27.29 | 32.13 | 45.00 | 45.15 |
| NMI | 50.19 | 52.33 | 80.59 | 84.15 | 17.16 | 17.11 | 46.32 | 51.08 | 27.32 | 26.73 |
| Purity | 51.99 | 54.41 | 88.75 | 90.70 | 23.98 | 24.42 | 32.28 | 37.64 | 65.48 | 65.67 |
| Time | 0.05 | 0.004 | 0.09 | 0.01 | 1.66 | 0.02 | 6.08 | 0.02 | 89.19 | 0.07 |

Table 8: Experimental results of the proposed method in comparison with the original MKKM.

| | Flo17 | | Digit | | CCV | | Flo102 | | Reuters | |
|---|---|---|---|---|---|---|---|---|---|---|
| | MKKM | Approx. | MKKM | Approx. | MKKM | Approx. | MKKM | Approx. | MKKM | Approx. |
| ACC | 45.37 | 46.69 | 47.00 | 47.55 | 18.01 | 17.75 | 21.96 | 23.26 | 44.99 | 45.12 |
| NMI | 45.35 | 46.28 | 48.16 | 48.62 | 15.22 | 13.81 | 42.30 | 43.14 | 27.29 | 26.73 |
| Purity | 46.84 | 48.82 | 49.70 | 50.15 | 22.25 | 21.97 | 27.61 | 27.68 | 65.50 | 65.66 |
| Time | 0.37 | 0.13 | 0.53 | 0.15 | 7.05 | 0.14 | 27.09 | 0.13 | 168.5 | 1.83 |

Table 9: Experimental results of the proposed method compared with the original MKKMMR.

| | Flo17 | | Digit | | CCV | | Flo102 | | Reuters | |
|---|---|---|---|---|---|---|---|---|---|---|
| | MKKMMR | Approx. | MKKMMR | Approx. | MKKMMR | Approx. | MKKMMR | Approx. | MKKMMR | Approx. |
| ACC | 58.01 | 53.82 | 90.90 | 89.45 | 22.37 | 23.45 | 40.13 | 33.70 | 45.70 | 47.61 |
| NMI | 55.47 | 54.44 | 83.70 | 81.91 | 18.62 | 19.17 | 57.27 | 50.75 | 27.64 | 29.01 |
| Purity | 59.04 | 55.51 | 90.90 | 89.45 | 25.66 | 25.99 | 46.39 | 39.38 | 65.72 | 67.04 |
| Time | 2.50 | 0.77 | 1.79 | 1.49 | 25.80 | 7.14 | 73.63 | 25.14 | 473.2 | 3.63 |