# OpenReview forum: "Stability and Generalization of Kernel Clustering: from Single Kernel to Multiple Kernel"
_NeurIPS.cc/2022/Conference — NeurIPS 2022 Accept_

### Official Review · Reviewer_YqyK · 2022-07-04

**Rating:** 7
**Confidence:** 5
**Soundness:** 4 excellent
**Presentation:** 3 good
**Contribution:** 3 good

**Summary:**

The paper proposes a new method for efficient multiple kernel clustering (MKC) over out-of-sample data. This method has the potential to scale the existing MKC algorithms to handle large-scale datasets. Then, the generalization ability of the method is studied from the single kernel setting to the multiple kernel setting, which provides valuable insight of the algorithm. Finally, numerical experiments are conducted to verify the superiority of the proposed algorithm.

**Questions:**

Please check the Weakness box.

**Limitations:**

None.

**Strengths And Weaknesses:**

Some strengths:
1. This work studies the generalization ability of the clustering algorithm based on algorithmic stability, which is a significant and interesting research topic.  By selecting $n$ samples as landmarks, the approximation embedding of the remaining samples can be computed with $O(N)$ time complexity, where $N$ is the sample number. Thus, the method can make existing MKC scalable to large-scale datasets.
2. The theoretical analysis of the paper is novel and sound. Theoretical analysis has proved that the proposed method has desirable generalization ability ($\mathcal{O}((m+1)Kn^{-3/2}+n^{-1/2})$), which provides a tighter bound than the one ($\mathcal{O}(mKn^{-1/2})$) provided in [1]. It guarantees the effectiveness of the proposed method. Moreover, the analysis can be scalable to multiple kernel clustering and the proofs are correct.
3. The paper is easy to follow and well-written.

Some weaknesses:
1. The paper only discusses uniform sampling as the selected method of landmarks. Other sampling strategies (e.g., leverage score [2]) could be tried to further improve the performance of the algorithm.
2. The authors haven’t analyzed the optimal number of landmarks. In the experiments, the number of landmarks is regarded as a hyper-parameter, which seems to require more effort to select the best one.
3. The authors only provide the theoretical and empirical results about SimpleMKKM. Some classic MKC algorithms are not discussed.

[1] Xinwang Liu, En Zhu, Jiyuan Liu, Timothy Hospedales, Yang Wang, and Meng Wang. Simplemkkm: Simple multiple kernel k-means. In arXiv preprint arXiv:2005.04975, 2020.

[2] Petros Drineas, Malik Magdon-Ismail, Michael W. Mahoney, and David P. Woodruff. Fast approximation of matrix coherence and statistical leverage. Journal of Machine Learning Research, 13:3441–3472, 2012.

---

> ### Author Response · Authors · 2022-07-31
> **Response to Reviewer YqyK**
>
> Thanks for your inspiring comments. The detailed replies are as follows.
>
> ---
>
> Q1: The paper only discusses uniform sampling as the selected method of landmarks. Other sampling strategies (e.g., leverage score [2]) could be tried to further improve the performance of the algorithm.
>
> A1: The methods such as leverage score sampling and ridge leverage score sampling have been used to improve the Nyström method in kernel $k$-means. However, there lacks relevant researches about the selection of landmarks in multiple kernel clustering (MKC). This is still an open problem in MKC. We will try to study the effect of sampling strategy of the proposed method in our future work.
>
> ---
>
> Q2: The authors haven’t analyzed the optimal number of landmarks. In the experiments, the number of landmarks is regarded as a hyper-parameter, which seems to require more effort to select the best one.
>
> A2: The authors in [1] give a theoretical research about the optimal number of landmarks of Nyström method for kernel $k$-means. The optimal number is given as $\mathcal{O}(\sqrt{n})$, where $n$ is the sample number. However, in our experiments, we find that $\mathcal{O}(\sqrt{n})$ may be too many for MKC. In most datasets, the used landmarks are far less than $\mathcal{O}(\sqrt{n})$. Because the empirical clustering risk will vary greatly with different kernel weights, it may be very difficult to study the optimal number of MKC in theory. We will try to address this issue in the future research.
>
> ---
>
> Q3: The authors only provide the theoretical and empirical results about SimpleMKKM. Some classic MKC algorithms are not discussed.
>
> A3: We conduct additional experiments on three classic algorithms, i.e., average multiple kernel $k$-means (AMKKM), multiple kernel $k$-means (MKKM) [3] and multiple kernel k-means clustering with matrix-induced regularization (MKKMMR) [2]. The results are reported in Section B5 of the appendix. As seen from the first two tables, our method achieve comparable clustering performance in the comparison with the standard AMKKM and MKKM, while the running time is far less. However, the results of our method fluctuates more dramatically when we apply it on MKKMMR. It is caused by two main reasons: 1) The hyper-parameter of MKKMMR makes the kernel weights be instable; 2) The optimal hyper-parameter of MKKMMR on landmarks is different to the whole training dataset. Through the experimental results and our empirical analysis, we find that our method would be more effective on the parameter-free multiple kernel clustering algorithms.
>
> ---
>
> [1] Calandriello D, Rosasco L. Statistical and computational trade-offs in kernel k-means. In NeurIPS 2018.
>
> [2] Liu X, Dou Y, Yin J, et al. Multiple kernel k-means clustering with matrix-induced regularization. In AAAI 2016.
>
> [3] Huang H C, Chuang Y Y, Chen C S. Multiple kernel fuzzy clustering. In TFS 2011.

---

> > ### Comment · Reviewer_YqyK · 2022-08-09
> > **Response to authors**
> >
> > Thanks for the responses. My concerns have been addressed, and I'd like to keep my initial rating.

---

### Official Review · Reviewer_Ymww · 2022-07-05

**Rating:** 7
**Confidence:** 4
**Soundness:** 4 excellent
**Presentation:** 3 good
**Contribution:** 3 good

**Summary:**

The paper provides a general method that enables multiple kernel clustering algorithms to compute the embedding of out-of-sample points efficiently. Instead of re-computing the sample embedding of the unseen samples with the support of the whole data matrix, the authors directly obtain the embedding by constructing the embedding functions through approximating the eigen function of the integral operator defined by the optimal kernel function. With these functions, the embedding of the out-of-sample data can be easily and accurately calculated. Moreover, the upper bound of excess clustering risk is established based on the studying of stability of the proposed method. Experimental results demonstrate the effectiveness and efficiency of the proposed method.

**Questions:**

1. Why do the authors select SimpleMKKM as a baseline for the implementation of the proposed method?

2. The proposed method for construction of the approximated indicator functions is inspired by [3]. However, the contribution over the mentioned method is not clearly described in the manuscript. More information should be given.

[3] Rosasco L, Belkin M, De Vito E., On learning with integral operators. In JMLR, page 905–934, 2010.

**Limitations:**

The limitation of the proposed method is considered in the conclusions. The societal impact is not discussed, and this will not cause any potential negative societal impact.

**Strengths And Weaknesses:**

Strengths

1. The out-of-sample problem of MKC studied by this paper is important and challenging. The idea of enabling kernel clustering algorithms to obtain the embedding of out-of-sample data is novel and interesting.

2. The conducted generalization study of clustering algorithms is an important theoretical contribution.
(1) The paper for the first time establishes the excess risk bound of kernel k-means by the analysis of its algorithmic stability.
(2)  The analysis is extended to multiple kernel setting. The proposed theoretical results enlighten a new way to study the generalization ability of multiple kernel clustering algorithms.

3. The paper is well written with clear and rational motivation. It presents a novel way of scaling the method to large scale multiple kernel clustering scenario and a new perspective of generalization analysis of MKC. These are valuable for AI researchers in this field.


Weakness

1. In the paper, only SimpleMKKM is used as an example to verify the effectiveness of the out-of-sample method. As a consequence, the experimental support of the paper is relatively weak. It will be better if more state-of-the-art multiple kernel clustering algorithms, such as [1][2], can be analyzed.

[1] Liu, X.; Dou, Y.; Yin, J.; Wang, L.; and Zhu, E. 2016. Multiple kernel k-means clustering with matrix-induced regularization. In AAAI, 1888–1894.

[2] Gonen, M., and Margolin, A. A. 2014. Localized data fusion for kernel k-means clustering with application to cancer biology. In NIPS, 1305–1313.

2. Some minor issues are as follows.

I. In the proof of Proposition 4.1, $R(H)$ should be written as $R(H,P)$ as its definition in the main text.

II. “Denote the k-th eigenvalue/vector pairs” in Line 483.

---

> ### Author Response · Authors · 2022-07-31
> **Response to Reviewer Ymww**
>
> Thanks a lot for your valuable feedback! We will answer your questions point by point.
>
> ---
>
> Q1: In the paper, only SimpleMKKM is used as an example to verify the effectiveness of the out-of-sample method. As a consequence, the experimental support of the paper is relatively weak. It will be better if more state-of-the-art multiple kernel clustering algorithms, such as [1][2], can be analyzed.
>
>
> A1: The algorithm in [2] needs to calculate the weight for each sample of each view, thus the proposed method can't be performed on it. We conduct additional experiments on three classic algorithms, i.e., average multiple kernel $k$-means (AMKKM), multiple kernel $k$-means (MKKM) [3] and multiple kernel k-means clustering with matrix-induced regularization (MKKMMR) [1]. The results are reported in Section B5 of the appendix. As seen from the first two tables, our method achieves comparable clustering performance in the comparison with the standard AMKKM and MKKM, while the running time is far less. However, the results of our method fluctuates more dramatically when we apply it on MKKMMR. It is caused by two main reasons: 1) The hyper-parameter of MKKMMR makes the kernel weights be instable; 2) The optimal hyper-parameter of MKKMMR on landmarks is different to the whole training dataset. Through the experimental results and our empirical analysis, we find that our method would be more effective on the parameter-free multiple kernel clustering algorithms.
>
> ---
>
> Q2: Some minor issues are in the manuscript.
>
> A2: Thank you for pointing out these issues. We have addressed them in the revised manuscript.
>
> ---
>
> Q3: Why do the authors select SimpleMKKM as a baseline for the implementation of the proposed method?
>
> A3: The reasons why we select SimpleMKKM as the baseline are as follows.
> 1) SimpleMKKM is one of the state-of-the-art algorithms which has promising clustering performance and efficiency. Moreover, SimpleMKKM has no hyper-parameter, and is more practical for application.
> 2) Through experimental observation, we find that the kernel weights of SimpleMKKM is stable against the training sample. In the theoretical analysis, because the optimization method is gradient descent, we can prove the stability of SimpleMKKM by studying the variation of kernel weights in each iteration. Other MKC algorithms may also have stability, but it's difficult to analyse in theory.
>
> ---
>
> Q4: The proposed method for construction of the approximated indicator functions is inspired by [4]. However, the contribution over the mentioned method is not clearly described in the manuscript. More information should be given.
>
> A4: We extend the method in [4] from single kernel to multiple kernel, and successfully apply it in the fields of clustering algorithms. Denote the integral operator $L_{\kappa}$ associated with the kernel function $\kappa$ as
> $
> (L_{k}f)(\mathbf{x}) = \int_{\mathcal{X}}k(\mathbf{x},\mathbf{y})f(\mathbf{y})d\rho(\mathbf{y}).
> $
> The empirical kernel matrix is $\frac{1}{n}\mathbf{K}$, whose element is $K_{ij} = \kappa(\mathbf{x_i},\mathbf{x_j})$.  $L_{\kappa}$ can be seen as the expected form of $\frac{1}{n}\mathbf{K}$.  $L_{\kappa}$ and $\frac{1}{n}\mathbf{K}$ operate on different space. To study the relation of above two operators, the authors in [4] define two operators $T_{\mathcal{H}},T_n: \mathcal{H} \rightarrow \mathcal{H}$ as
> $
> T_{\mathcal{H}} = \int_{\mathcal{X}} \langle\cdot,\kappa_\mathbf{x} \rangle \kappa_\mathbf{x} d\rho(\mathbf{x})$ and $T_{n} = \frac{1}{n} \sum_{i=1}^n \langle\cdot,\kappa_{\mathbf{x_i}} \rangle \kappa_{\mathbf{x_i}},
> $
> where $\kappa_\mathbf{x} = \kappa(\cdot,\mathbf{x})$ and $\kappa_{\mathbf{x_i}} = \kappa(\cdot,{\mathbf{x_i}})$. By Proposition 9 of [4], the eigenvector and eigenfunction of $\frac{1}{n}\mathbf{K}$ and $T_n$ are as follows:
> $
> \mathbf{h_k} = \frac{1}{\sqrt{\lambda_k}} (u_k(\mathbf{x_1}),\cdots,u_k(\mathbf{x_n})), \quad u_k(\mathbf{x}) = \frac{1}{\sqrt{\lambda_k}} \left(\frac{1}{n} \sum_{i=1}^n h_{ik}\kappa(\mathbf{x},\mathbf{x_i}) \right),
> $
> where $\lambda_k$ is the $k$-th eigenvalue of $\frac{1}{n}\mathbf{K}$. It is easy to check that for any point $\mathbf{x}$ on the sample space, the embedding vector can be approximated as $[h_1(\mathbf{x}),\cdots,h_K(\mathbf{x})]$, where $h_k(\mathbf{x}) = \frac{1}{n\lambda_k} \left( \sum_{i=1}^n h_{ik}\kappa(\mathbf{x},\mathbf{x_i}) \right)$. In the setting of multiple kernel, kernel function is $\kappa_{\boldsymbol{\alpha}}$, thus we can approximate the embedding of $\mathbf{x}$ by Eq.(8) of the manuscript.
>
> ---
>
> [1] Liu X, Dou Y, Yin J, et al. Multiple kernel k-means clustering with matrix-induced regularization. In AAAI 2016.
>
> [2] Gönen M, Margolin A A. Localized data fusion for kernel k-means clustering with application to cancer biology. In NeurIPS 2014.
>
> [3] Huang H C, Chuang Y Y, Chen C S. Multiple kernel fuzzy clustering. In TFS 2011.
>
> [4] Rosasco L, Belkin M, De Vito E. On learning with integral operators. In JMLR 2010.

---

> ### Comment · Area_Chair_o8Kh · 2022-08-09
> **Author rebuttal phase closing today**
>
> The author-rebuttal phase closes today. Please acknowledge the author rebuttal and state if your position has changed. Thanks!

---

### Official Review · Reviewer_LK5c · 2022-07-09

**Rating:** 6
**Confidence:** 4
**Soundness:** 3 good
**Presentation:** 3 good
**Contribution:** 3 good

**Summary:**

The paper studies the problem of clustering out-of-sample data over multiple kernel clustering dataset. The authors of the paper seek to solve the problem by learning the embedding functions with a novel approximation strategy by take advantage of the relation between empirical kernel matrix and integral operator of the kernel function. Then, the generalization ability of the method is studied with valuable depth from single kernel setting to multiple kernel setting. Finally, numerical experiments are conducted to verify the superiority of the proposed algorithm.

**Questions:**

In experiments, the clustering performance of SimpleMKKM with the proposed method on some datasets is superior to the original one. I think that the authors should give more discussions about this phenomenon.

**Limitations:**

None.

**Strengths And Weaknesses:**

Strengths:
1. The proposed method can greatly accelerate MKC on handling out-of-sample problem. Though the method is simple, it can reduce the time complexity from $\mathcal{O}(N^3)$ to $\mathcal{O}(N)$. Consequently, the method can enable MKC to be performed on large-scale dataset.
2. The authors offer theoretical analysis about the proposed method. It is the first time the stability of clustering algorithm is analyzed. The paper is technically sound, and the proofs are strict.
3. The paper establishes a connection between the stability of MKC algorithm and the output kernel weights, which enlightens a design principle for this type of algorithms.
Weakness:
1. Some other methods for large-scale implementation of kernel clustering algorithms should be referred, e.g., Nystr\"om [1] and random Fourier feature [2].

2. The proposed clustering risk bound should be compared with other relevant works.

[1] Cameron Musco and Christopher Musco. Recursive sampling for the Nystr\"om method. In NeurIPS 2017.

[2] Radha Chitta, Rong Jin, and Anil K. Jain. Efficient Kernel Clustering Using Random Fourier Features. In ICDM 2012.

3. Experiments are not sufficient. In Section 5.1, the authors provide theoretical analysis of clustering on single kernel. However, the paper lacks corresponding experiments.

4. The codes of experiments are not available.

---

> ### Author Response · Authors · 2022-07-31
> **Response to Reviewer LK5c**
>
> Thanks for constructive comments to improve the paper.  We address the points you raised one by one.
>
> ---
>
> Q1: Some other methods for large-scale implementation of kernel clustering algorithms should be referred, e.g., Nyström [1] and random Fourier feature [2].
>
> A1: Thanks, we discuss the mentioned methods as follows. "To improve the scalability of kernel clustering algorithms, methods such as Nyström [1] approximation and random Fourier feature (RFF) [2] are proposed. These two methods acquire the non-linear feature of samples in real space through the approximating the kernel matrix. However, Nyström method can't be implemented on out-of-sample points directly. RFF can fill this gap, but the dimension of the learned embedding is comparably large such that the subsequent clustering process is time consuming. More seriously, because it's difficult to bound the difference of the kernel weights before and after the approximation, these two methods are rarely implemented on MKC algorithms." We will add the above contents to the introduction section to make the point clear.
>
> ---
>
> Q2: The proposed clustering risk bound should be compared with other relevant works.
>
> A2: In existing literature, the authors in [3] propose the excess risk bound of kernel $k$-means as $\mathcal{O}\left(\frac{K}{\sqrt{n}}\right)$, where $K$ is the cluster number and $n$ is the number of samples. As an improvement of the above result, the authors in [4] bound the excess risk as $\mathcal{O}\left(\sqrt{\frac{K}{n}}\right)$, which matches with the
> stated lower bound and is nearly optimal. In this paper, the proposed excess risk bound of single kernel is $\widetilde{\mathcal{O}}\left(\frac{K}{n\sqrt{n}}+\frac{1}{\sqrt{n}}\right)$ with two mild assumptions, which is tighter than $\mathcal{O}\left(\sqrt{\frac{K}{n}}\right)$. In multiple kernel clustering, the proposed bound is $\tilde{\mathcal{O}}\left(\frac{(m+1)K}{n\sqrt{n}}+\frac{1}{\sqrt{n}}\right)$, which is tighter than the existing results $\mathcal{O}(\frac{mK}{\sqrt{n}})$ proposed in [5].
>
> ---
>
> Q3: Experiments are not sufficient. In Section 5.1, the authors provide theoretical analysis of clustering on single kernel. However, the paper lacks corresponding experiments.
>
>
> A3: To give empirical evidence of our theoretical results, we conduct experiments on single kernel, and the results are reported in Section B5 of the revised manuscript. All experimental setting and datasets are the same as Section 6.1. The used kernel matrix is constructed by the average summation of all base kernel matrices. As seen from the Table 7 in the appendix, we can find that our method achieve comparable clustering performance in comparison with the standard kernel $k$-means. As stated in Section 4.2, the proposed method sharply reduces the running time. It shows that our method is effective to clustering algorithm on single kernel, i.e., kernel $k$-means.
>
> ---
>
> Q4: The codes of experiments are not available.
>
> A4: The code is now available in an anonymous website. Please refer to https://anonymous.4open.science/r/MKC_approximation-6091/.
>
> ---
>
> Q5: In experiments, the clustering performance of SimpleMKKM with the proposed method on some datasets is superior to the original one. I think that the authors should give more discussions about this phenomenon.
>
> A5: Our method aims to approximate the exact embedding $\mathbf{H}$ which is obtained by original multiple kernel clustering (MKC) algorithm. As shown in [5], when the objective value of original MKC algorithm converges to optima, the exact $\mathbf{H}$ may not obtain the best clustering performance. Thus, in some datasets, the approximation embedding can obtain better but similar clustering performance comparing with the exact one.
>
> ---
>
> [1] Musco C, Musco C. Recursive sampling for the nyström method. In NeurIPS 2017.
>
> [2] Chitta R, Jin R, Jain A K. Efficient kernel clustering using random fourier features. In ICDM 2012.
>
> [3] Biau G, Devroye L, Lugosi G. On the performance of clustering in Hilbert spaces. In TIT 2008.
>
> [4] Liu Y. Refined Learning Bounds for Kernel and Approximate $ k $-Means. In NeurIPS 2021.
>
> [5] Liu X, Zhu E, Liu J, et al. SimpleMKKM: Simple multiple kernel k-means. arXiv preprint 2020.

---

> > ### Comment · Reviewer_LK5c · 2022-08-10
> > **Respond to rebuttal**
> >
> > Thanks for the explanations. I have read the author's feedback. I would like to keep my review unchanged, and continue to support acceptance for this paper!

---

> ### Comment · Area_Chair_o8Kh · 2022-08-09
> **Author rebuttal phase closing**
>
> The author-rebuttal phase closes today. Please acknowledge the author rebuttal and state if your position has changed. Thanks!

---

### Meta-Review · Area_Chair_o8Kh · 2022-08-28

**Recommendation:** Accept
**Confidence:** Certain

**Metareview:**

The paper introduces a methodology for clustering out-of-sample data in the multiple kernel clustering (MKC) problem by leveraging the relationship between the empirical kernel matrix and the integral operator of the kernel function. Clustering risk bounds for the proposed method are provided that compare favorably with the literature, and numerical experimentation shows that the methodology performs well when applied to algorithms developed for both single and multiple kernel clustering. The reviewers concur that the methodology enables efficient large-scale MKC and provides a novel perspective on its generalization analysis.

**Award:**

No

---

### Decision · Program_Chairs · 2022-09-14

Accept